# Label-Only Model Inversion Attacks via Knowledge Transfer

**Ngoc-Bao Nguyen**[*1]     **Keshigeyan Chandrasegaran**[*2‡]
**Milad Abdollahzadeh**[1]     **Ngai-Man Cheung**[1†]
[1]Singapore University of Technology and Design (SUTD)     [2]Stanford University
`thibaongoc_nguyen@mymail.sutd.edu.sg  ngaiman_cheung@sutd.edu.sg`

## Abstract

In a model inversion (MI) attack, an adversary abuses access to a machine learning (ML) model to infer and reconstruct private training data. Remarkable progress has been made in the white-box and black-box setups, where the adversary has access to the complete model or the model's soft output respectively. However, there is very limited study in the most challenging but practically important setup: Label-only MI attacks, where the adversary only has access to the model's predicted label (hard label) without confidence scores nor any other model information.

In this work, we propose LOKT, a novel approach for label-only MI attacks. Our idea is based on transfer of knowledge from the opaque target model to surrogate models. Subsequently, using these surrogate models, our approach can harness advanced white-box attacks. We propose knowledge transfer based on generative modelling, and introduce a new model, Target model-assisted ACGAN (T-ACGAN), for effective knowledge transfer. Our method casts the challenging label-only MI into the more tractable white-box setup. We provide analysis to support that surrogate models based on our approach serve as effective proxies for the target model for MI. Our experiments show that our method significantly **outperforms existing SOTA Label-only MI attack by more than 15% across all MI benchmarks.** Furthermore, our method compares favorably in terms of query budget. Our study highlights rising privacy threats for ML models even when minimal information (i.e., hard labels) is exposed. Our code, demo, models and reconstructed data are available at our project page: https://ngoc-nguyen-0.github.io/lokt/

## 1   Introduction

Model inversion (MI) attacks aim to infer and reconstruct sensitive private samples used in the training of models. MI and their privacy implications have attracted considerable attention recently [1, 2, 3, 4, 5, 6, 7, 8, 9, 10, 11]. The model subject to MI is referred to as *target model*. There are three categories of MI attacks: (1) White-box MI, where complete target model information is accessible by the adversary [1, 2, 3, 5, 7, 10]; (2) Black-box MI, where target model's soft labels are accessible [4, 12, 10, 13]; (3) Label-only MI, where only target model's hard labels are accessible [6]. This paper focuses on label-only MI, which is the most challenging setup as only limited information (hard labels) is available (Fig. 1).

In most existing work, MI attack is formulated as an optimization problem to seek reconstructions that maximize the likelihood under the target model [1, 2, 3, 6]. For DNNs, the optimization problems are highly non-linear. When the sensitive private samples are high-dimensional samples (e.g. face

---

[*] These authors contributed equally.     [‡] Work done while at SUTD.

[†] Corresponding author.

images), the optimizations are ill-posed, even in white-box setups. To overcome such issues, recent MI [1, 2, 3, 5, 10, 7, 6, 11] learn distributional priors from public data via GANs [14, 15, 16, 17], and solve the optimization problems over GAN latent space rather than the unconstrained image space. For example, MI attacks on face recognition systems could leverage GANs to learn face manifolds from public face images which have no identity intersection with private training images. White-box attacks based on public data and GANs have achieved remarkable success [1, 2, 3, 7, 11]. We follow existing work and recent label-only MI [6] and leverage public data in our method. Furthermore, similar to existing work, we use face recognition models as examples of target models.

**Research gap.** Different from white-box attack, study on label-only attack is limited despite its practical importance, e.g., many practical ML models only expose predicted labels. Focusing on label-only attack and with no knowledge of internal workings of target model nor its confidence score, BREPMI [6] takes a *black-box search* approach to explore the search space iteratively (Fig. 1(a)). To seek reconstructions with high likelihood under target model, [6] proposes to query target model and observe the model's hard label predictions, and update search directions using *Boundary Repelling* in order to move towards centers of decision regions, where high likelihood reconstructions could be found. However, black-box search in the high-dimensional latent space is extremely challenging.

**In this paper**, we propose a new approach for Label-Only MI attack using Knowledge Transfer (LOKT). Instead of performing a black-box search approach as demonstrated in [6] and directly searching high-likelihood reconstruction from the opaque target model (Fig. 1(a)), which could be particularly challenging for high-dimensional search space, we propose a different approach. Our approach aims to transfer the decision knowledge of the target model to surrogate models, for which complete model information is accessible. Subsequently, with these surrogate models, we could harness SOTA white box attacks to seek high-likelihood reconstructions (Fig. 1(b)). To obtain the surrogate models, we explore generative modeling [18, 19, 20, 21, 22]. In particular, we propose a new Target model-assisted ACGAN, T-ACGAN, which extends ACGAN [23] and leverages our unique problem setup where we have access to the predicted labels of the target model as shown in Fig. 1(d). In particular, by effectively leveraging the target model in discriminator/classifier training, we can explore *synthetic data* for decision knowledge transfer from the target model to the surrogate model. With T-ACGAN capturing the data manifold of public samples, synthetic data is diverse and abundant. We hypothesize that such rich synthetic data could lead to improved decision knowledge transfer. Moreover, as training progresses, T-ACGAN generator learns to improve its conditional generative capabilities, enabling it to produce more balanced synthetic data for surrogate model learning. We explore several surrogate model designs. In one configuration, we employ the discriminator/ classifier of T-ACGAN as the surrogate model. In an alternative design, we utilize the generator of T-ACGAN to train different surrogate model variants. It's noteworthy that the generator of T-ACGAN can be readily employed for white-box attacks, and its conditional generation capabilities can effectively reduce the search space during inversion. **In addition, we perform analysis to support that our surrogate models are effective proxies for the opaque target model for MI.** (Fig. 1(e)). Overall, our T-ACGAN renders improved surrogate models, resulting in a significant boost in MI attack accuracy (Fig. 1(f)) and reduced number of queries compared to previous SOTA approach. **Our contributions are:**

- We propose LOKT, a new label-only MI by transferring decision knowledge from the target model to surrogate models and performing white-box attacks on the surrogate models (Sec. 4). Our proposed approach is the first to address label-only MI via white-box attacks.

- We propose a new T-ACGAN to leverage generative modeling and the target model for effective knowledge transfer (Sec. 4).

- We perform analysis to support that our surrogate models are effective proxies for the target model for MI (Sec. 5).

- We conduct extensive experiments and ablation to support our claims. Experimental results show that our approach can achieve significant improvement compared to existing SOTA MI attacks (Sec. 6). Additional experiments/ analysis are in Supplementary.

## 2 Related work

Model Inversion (MI) has particularly alarming consequences in security-sensitive domains, such as face recognition [24, 25, 26, 27], medical diagnosis [28, 29, 30]. Fredrikson et al. [31] introduces the

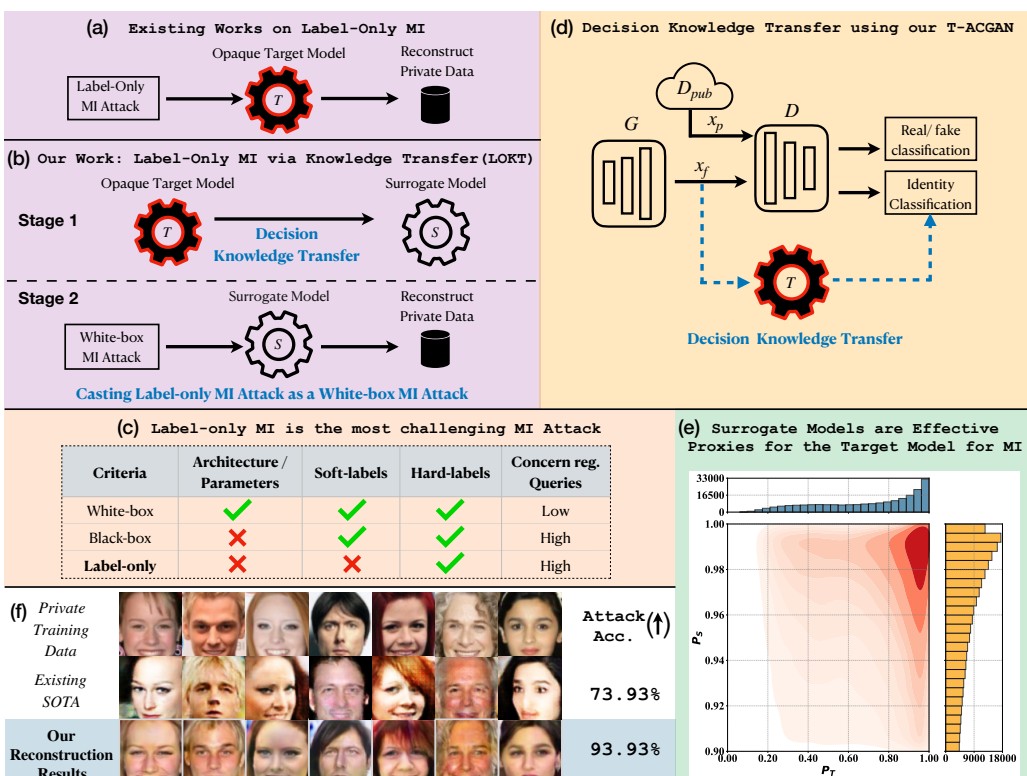

Figure 1: *Overview and our contributions.* **(a)** Under Label-only model inversion (MI) attack, the Target model $T$ is opaque. **(b) Stage 1:** As our first contribution, we propose a knowledge transfer scheme to render surrogate model(s). **(b) Stage 2:** Then, we cast the Label-only MI attack as a white-box MI attack on surrogate model(s) $S$. **(c)** This casting can ease the challenging problem setup of label-only MI attack into a white-box MI attack. To our knowledge, our proposed approach is the first to address label-only MI via white-box MI attacks. **(d)** We propose T-ACGAN to leverage generative modeling and the target model for effective knowledge transfer to render surrogate model(s). Knowledge transfer renders $D$ (Discriminator) as a surrogate model, and further generated samples of T-ACGAN can be used to train additional surrogate variant $S$ (Sec. 4.3). **(e)** Our analysis demonstrates that $S$ is an effective proxy for $T$ for MI attack (details in Sec.5). In particular, white-box MI attack on $S$ mimics the white-box attack on opaque $T$. **(f)** Our proposed approach significantly improves the Label-only MI attack (e.g. $\approx 20\%$ improvement in standard CelebA benchmark compared to existing SOTA [6]) resulting in significant improvement in private data reconstruction. Best viewed in color.

first MI attack for simple linear regression models. Recently, several works extend MI for complex DNNs under different setups. For white-box setup, [1] proposes Generative Model Inversion (GMI) to leverage public data and GAN[32, 33] to constrain the search space. [2] proposes Knowledge-Enriched Distributional Model Inversion (KEDMI) to train an inversion-specific GAN for the attack. [3] proposes Variational Model Inversion (VMI) to apply variational objectives for the attack. Very recent work [7] proposes Pseudo Label-Guided MI (PLG-MI) to apply model's soft output to train a conditional GAN (cGAN)[34] for white-box attack. LOMMA[11] proposes a better objective function for MI and model augmentation to address MI overfitting. For black-box attack, where model's soft output is available, [4] proposes to train an inversion model and a decoder to generate target images using predicted scores of the inversion model. [12] proposes an adversarial approach for black-box MI. For label-only attack, [6] proposes BREPMI, the first label-only MI using a black-box Boundary Repelling search. See Supplementary for further discussion of related work.

## 3 Problem setup

Given a target model $T$, the goal of MI is to infer private training data $\mathcal{D}_{priv}$ by abusing access to model $T$. More specifically, given a target class/ identity label $y$, the adversary aims to reconstruct

an image $x$ which is similar to the images of class $y$ in $\mathcal{D}_{priv}$. Most MI formulate the inversion as optimization problems to seek the highest likelihood reconstructions for identity $y$ under $T$. As direct searching for $x$ in the unconstrained image space is ill-posed, many MI attacks [1, 2, 3, 7, 6] leverage public dataset $\mathcal{D}_{pub}$ that is the same domain as $\mathcal{D}_{priv}$, e.g., $\mathcal{D}_{priv}$ and $\mathcal{D}_{pub}$ are facial image datasets. GAN [14] is applied to learn distributional prior from $\mathcal{D}_{pub}$, and the adversary searches the GAN latent space instead of the unconstrained image space for high-likelihood reconstructions under $T$:

$$\max_z \log P_T(y|G(z)) \tag{1}$$

Here, $G$ is the generator, and $P_T(y|.)$ is the likelihood of an input for identity $y$ under target model $T$. White-box attacks apply gradient ascent and some regularization [1, 2, 3, 7] to solve Eq. 1, whereas label-only attack BREPMI [6] applies black-box search to tackle Eq. 1. In this paper, we also tackle Eq. 1 under label-only setup, i.e. only the predicted label is available.

## 4 Approach

Our proposed label-only MI consists of two stages. In stage 1, we learn surrogate models. In stage 2, we apply SOTA white-box attack on the surrogate models. To learn surrogate models, we explore an approach based on GAN and propose a new Target model-assisted ACGAN (T-ACGAN) for effective transfer of decision knowledge. Our T-ACGAN learns the generator $G$ and the discriminator $D$ with classifier head $C$. In one setup, we directly take $C \circ D$ as the surrogate model*. In another setup, we apply $G$ to generate synthetic data to train another surrogate model $S$ or an ensemble of $S$. Then, we apply SOTA white-box attack on $C \circ D$, $S$ or the ensemble of $S$. In our experiments, we show that using $C \circ D$ in a white-box attack can already outperform existing SOTA label-only attack. Using $S$ or an ensemble of $S$ can further improve attack performance. The $G$ obtained from our T-ACGAN can be readily leveraged in the attack stage.

### 4.1 Baseline

Before discussing our proposed approach, we first discuss a simple baseline for comparison. Given the public data, one could directly use the target model $T$ to label the data and learn the surrogate model $S$. For $x_p \in \mathcal{D}_{pub}$, we construct $(x_p, \tilde{y})$, where $\tilde{y} = T(x_p)$ is pseudo label of *private* identity. We obtain the dataset $\tilde{\mathcal{D}}_{pub}$ with samples $(x_p, \tilde{y})$, i.e. $\tilde{\mathcal{D}}_{pub}$ is the public dataset with pseudo labels. We apply $\tilde{\mathcal{D}}_{pub}$ to train $S$. However, this algorithm suffers from class imbalance. In particular, some private identities could have less resemblance to $x_p \in \mathcal{D}_{pub}$. As a result, for some $\tilde{y}$, there is only a small number of $x_p$ classified into it, and $\tilde{\mathcal{D}}_{pub}$ is class imbalanced. When using $\tilde{\mathcal{D}}_{pub}$ to train $S$, minority classes may not gain adequate decision knowledge under $T$ and could perform sub-optimally. In our experiments, we also apply techniques to mitigate the class imbalance in $\tilde{\mathcal{D}}_{pub}$. However, the performance of this baseline approach is inadequate as we will show in the experiments.

### 4.2 Review of ACGAN

In standard ACGAN [23], we are given a real training dataset with label, i.e., $\mathcal{D}_{real}$ with samples $(x_r, y)$. The generator $G$ takes a random noise vector $z$ and a class label $y$ as inputs to generate a fake sample $x_f$. The discriminator $D$ outputs both a probability distribution over sources $P(s|x) = D(x)$, where $s \in \{Real, Fake\}$, and a probability distribution over the class labels, i.e., $P(c|x) = C \circ D(x)$, and $c$ is one of the classes. For real training sample $x_r$ of label $y$ and fake sample $x_f = G(z, y)$ with conditional information $y$, the loss functions for $D$, $C$ and $G$ are:

$$\begin{aligned}\mathcal{L}_{D,C} = &-E[\log P(s = Fake|x_f)] - E[\log P(s = Real|x_r)] \\ &- E[\log P(c = y|x_f)] - E[\log P(c = y|x_r)]\end{aligned} \tag{2}$$

$$\mathcal{L}_G = E[\log P(s = Fake|x_f)] - E[\log P(c = y|x_f)] \tag{3}$$

---

*With a slight abuse of notation we use $D$ to represent the entire discriminator and the discriminator up to and including the penultimate layer in the context of $C \circ D$.

### 4.3 Our Proposed T-ACGAN and Learning of Surrogate Model

Unlike standard ACGAN setup where we have access to labelled data $\mathcal{D}_{real}$ with samples $(x_r, y)$, in our setup, we have access to real public data without label: $\mathcal{D}_{pub}$ with samples $x_p$. *Importantly, we can leverage the target model $T$ to provide pseudo labels for generated samples $x_f = G(z, y)$, which are diverse and abundant.* Our proposed T-ACGAN aims to take advantage of $T$ to provide more diverse and accurate pseudo labelled samples during the training.

**$D$ and $C$ Learning.** Our T-ACGAN leverages $T$ to assign pseudo labels to the diverse generated samples $x_f = G(z, y)$, i.e., $\tilde{y} = T(x_f)$. We apply samples $x_p$ and $(x_f, \tilde{y})$ to learn $D$ and $C$:

$$\mathcal{L}_{D,C} = -E[\log P(s = Fake|x_f)] - E[\log P(s = Real|x_p)] \\ - E[\log P(c = \tilde{y}|x_f)] \tag{4}$$

In Eq. 4, the term $E[\log P(c = \tilde{y}|x_f)] = E[\log P(c = \tilde{y}|G(z, y))]$ is different from ACGAN and may look intriguing. Instead of using $y$ as class supervision to train $D$ and $C$ as in ACGAN (Eq. 2), our T-ACGAN takes advantage of $T$ to apply $\tilde{y} = T(G(z, y))$ to train $D$ and $C$, as $\tilde{y}$ is more accurate conditional information compared with $y$ especially during the initial epochs. *With Eq. 4, our method transfers the decision knowledge of $T$ into $D$ and $C$ via diverse generated samples.* Furthermore, as we can generate diverse pseudo labelled samples $(x_f, \tilde{y})$ using $T$ and $G$, pseudo labelled data based on $x_p$ can be omitted. In our experiment, we show that we can achieve good performance using diverse samples $(x_f, \tilde{y})$. In T-ACGAN, we utilize public data $x_p$ only for real/fake discrimination.

**$G$ Learning.** We follow ACGAN training for $G$, i.e. Eq. 3. With $D$ and $C$ trained with decision knowledge of $T$ in the above step, they provide feedbacks to $G$ to improve its conditional generation *in the private label space of $T$*. In our experiment, we analyze $y$ in $x_f = G(z, y)$ and $\tilde{y} = T(x_f)$. As training progresses, $G$ improves its conditional generation, and $y$ and $\tilde{y}$ become more aligned. Note that, as $T$ outputs only hard labels, $T$ cannot be readily applied to provide feedback for $G$ learning.

**Surrogate Model.** With alternating $D$ and $C$ learning and $G$ learning, we obtain $D$, $C$ and $G$. We explore three methods to obtain the surrogate model. ● (i) We directly take $C \circ D$ in T-ACGAN as the surrogate model and apply a white-box attack on $C \circ D$. This can be justified as $C \circ D$ is trained based on decision knowledge of $T$ to classify a sample into identities of private training data. ● (ii) We apply $G$ of T-ACGAN to generate dataset $\tilde{\mathcal{D}}_{fake}$ with samples $(x_f, \tilde{y})$, where $x_f = G(z, y)$ and $\tilde{y} = T(x_f)$. We apply $\tilde{\mathcal{D}}_{fake}$ to train another surrogate model $S$. ● (iii) We use the same dataset $\tilde{\mathcal{D}}_{fake}$ in (ii) to train an ensemble of $S$ of different architectures. As pointed out in [11], using an ensemble of $S$ could improve white-box attack performance.

**White-box Attack.** With surrogate model $C \circ D$, $S$ or an ensemble of $S$, any white-box attack can be applied. In our experiments, we show that our surrogate models are effective across a range of white-box attacks (See the Supplementary). Furthermore, $G$ in T-ACGAN can be readily leveraged for performing the attack. Particularly, based on $G(z, y)$ obtained in the above steps, we could reduce the search space during inversion to the latent region corresponding to the target identity $y$, leading to more efficient search and improved attack accuracy [7].

## 5 Analysis for justification of surrogate models

In this section, we provide an analysis to justify why our surrogate model could be an effective proxy for $T$ under MI, i.e., *the results of white-box MI attack on our surrogate model be good approximation to that of white-box MI attack on $T$*. Note that results of white-box MI on $T$ cannot be obtained directly as $T$ exposes only hard labels. To simplify the presentation, we focus our discussion on $S$. As discussed in Sec. 3, most MI attacks formulate inversion as an optimization problem of seeking reconstructions that achieve highest likelihood under target model. Therefore, when we carry out MI on $S$ with SOTA white-box approaches, we expect to obtain high-likelihood reconstructions under $S$ (or high-likelihood generated samples of GAN under $S$, see Eq. 1). We use $P_S$ and $P_T$ to denote likelihood of a sample under $S$ and $T$ respectively.

In what follows, we provide analysis to support that $S$ based on our approach would possess an important property of good proxy for $T$. **Property P1:** *For high-likelihood samples under $S$, it is likely that they also have high likelihood under $T$.* See Fig. 1(e) for distribution of generated samples' $P_T$ conditioning on those with high $P_S$. It can be observed that many have high $P_T$. Particularly, it is

uncommon for high-likelihood samples under $S$ to have low likelihood under $T$ (see Fig. 1(e) only a few samples have low $P_T$).

With **Property P1**, the result obtained by white-box on $S$ (which is a high likelihood sample under $S$) is likely to have a high likelihood under $T$ and could be a good approximation to the result of white-box on $T$ (which is a high likelihood sample under $T$). In Fig. 1(e), **P1** can be clearly observed[†]. Therefore, $S$ using our approach would possess **P1** and would be a good proxy for $T$ for MI.

**Why would $S$ possess property P1?** This could be intriguing. After all, $T$ does not expose any likelihood information. The labels of samples assigned by $T$ are the only information available to $S$ during training of $S$. It does not appear that $S$ can discern low or high-likelihood samples under $T$.

To discuss why $S$ would possess **P1**, we apply findings and analysis framework of Arpit et al. [35] regarding the general learning dynamics of DNNs. [35] presents a data-centric study of DNN learning with SGD-variants. In [35], "easy samples" are ones that fit better some patterns in the data (and correspondingly "hard samples"). The easy and hard samples exhibit high and low likelihoods in DNNs resp. as discussed in [35]. Furthermore, an important finding from [35] is that, in DNNs learning, the models learn simple and general patterns of the data *first* in the training stage to fit the easy samples.

We apply the framework of [35] to understand our learning of $S$ and the reason why $S$ would possess **P1**. Fig. 2(a) illustrates easy and hard samples in our problem: patterns of face identities can be observed in some samples (easy samples), while other samples (hard samples) exhibit diverse appearance. Similar to [35], Fig. 2(b) shows that these easy and hard face samples tend to have high and low likelihood under $T$. Fig. 2(c) shows the learning of $S$ on these easy and hard samples at different epochs. Consistent with the *"DNNs Learn Patterns First"* finding in [35], $S$ learns general identity patterns first to fit the easy samples. Therefore, $P_S$ of easy samples improve at a faster pace in the training, and many of them achieve high $P_S$. As easy samples tend to have high $P_T$, we observe **P1** in $S$. For the hard samples (which tend to have low $P_T$), it is uncommon for $S$ to achieve high likelihood on them as they do not fit easily to the pattern learned by $S$.

# 6    Experiments

In this section, we present extensive experiment results and ablation studies: (i) We show that our proposed T-ACGAN can lead to better surrogate models compared to alternative approaches (Sec. 6.2). (ii) We show that our proposed approach LOKT can significantly outperform the existing SOTA label-only MI attack (Sec. 6.3). (iii) We present additional results (Sec. 6.4) to demonstrate the efficacy of LOKT against SOTA MI defense methods. We further show that LOKT compares favorably in terms of query budget compared to existing SOTA. **Additional experiments/analysis provided in Supplementary.**

## 6.1    Experimental Setup

To ensure a fair comparison, we adopt the exact experimental setup used in BREPMI [6]. In what follows, we provide details of the experimental setup.

**Dataset.** We use three datasets, namely CelebA [40], Facescrub [41], and Pubfig83 [42]. We further study Label-Only MI attacks under distribution shift using FFHQ dataset [43] which contains images that vary in terms of background, ethnicity, and age. Following [2, 6], we divide each dataset (CelebA/ Facescrub/ Pubfig83) into two non-overlapping sets: private set $\mathcal{D}_{priv}$ for training the target model $T$, and public set $\mathcal{D}_{pub}$ for training GAN/T-ACGAN. More details on datasets and attacked identities can be found in Supplementary.

Table 1: Details of target model $T$. To showcase the effectiveness of our proposed method, we conduct a comprehensive set of 30 experiments, covering 10 different setups.

| $\mathcal{D}_{priv}$ | $T$ | |
|---|---|---|
| | **Architecture** | **# classes** |
| CelebA | FaceNet64 [36] | |
| | IR152 [37] | |
| | VGG16 [38] | 1,000 |
| | BiDO-HSIC [39] | |
| | MID [9] | |
| Facescrub | FaceNet64 [36] | 200 |
| Pubfig83 | FaceNet64 | 50 |

**Target Models.** Following [2, 6], we use 3 target models $T$ including VGG16 [38], IR152 [37], and FaceNet64 [36]. All target models are provided in [2, 6].

---

[†]Fig. 1(e) are $P_T$ and $P_S$ of $x_f = G(z, y)$ from our T-ACGAN. More details in Supp.

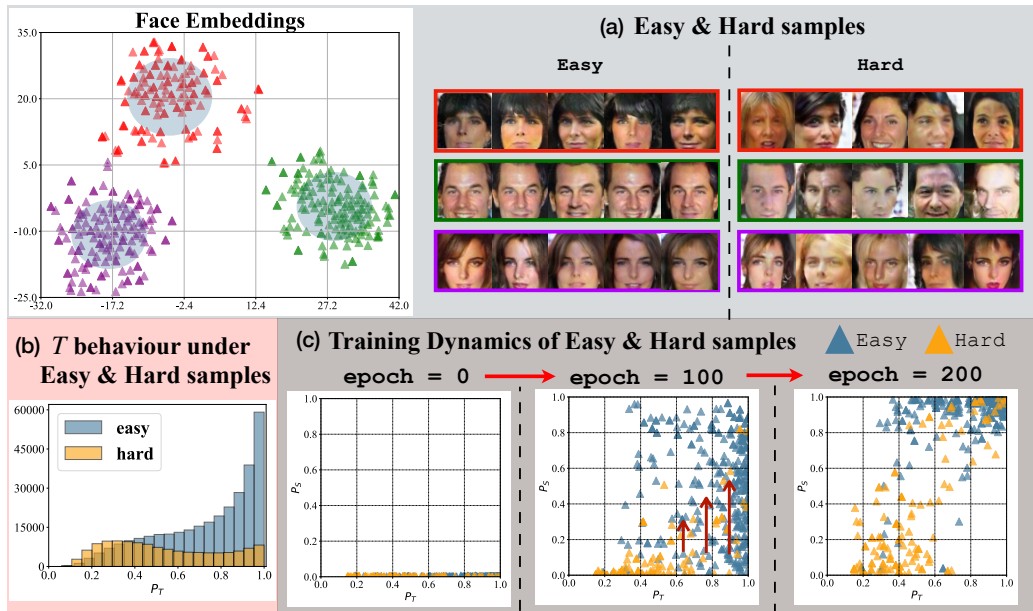

Figure 2: We apply the framework of [35] to analyze learning dynamics of $S$ to reason why $S$ possesses property P1, and therefore could be an effective proxy for $T$ under MI. We analyze generated samples $x_f$ from our T-ACGAN for 3 identities (IDs 20, 16, 36). Note that $x_f$ analysis is relevant as generated samples are used in MI attacks. **(a)**: We analyze face embeddings of $x_f$ extracted from publicly available SOTA face recognition model here. Different clusters and different distances from cluster centroids can be observed, suggesting patterns of face identities in some samples (easy samples) while diverse appearance in other samples (hard samples). We use distances from centroids to identify easy samples $x_f^e$ and hard samples $x_f^h$ (easy samples are indicated using transparent blue circle for each ID in the visualization). Visualization of $x_f^e$ and $x_f^h$ in image space further demonstrates identity patterns in $x_f^e$ and diverse appearance in $x_f^h$. **(b)**: Similar to [35], we observe that $x_f^e$ and $x_f^h$ tend to have high and low likelihood under $T$ ($P_T$) resp (training data). **(c)**: We track likelihood under $S$ ($P_S$) for $x_f^e$ and $x_f^h$ during the training of $S$. As training progresses, $P_S$ of $x_f^e$ and $x_f^h$ improve, and samples move up vertically (note that $P_T$ of samples do not change). Consistent with the *"DNNs Learn Patterns First"* finding in [35], $S$ learns general identity patterns first to fit the easy samples. Therefore, $P_S$ of $x_f^e$ improve at a faster pace in the training, and many of them achieve high $P_S$ at epoch = 200. As $x_f^e$ tend to have high $P_T$, we observe property P1 in $S$. For $x_f^h$ (many of them tend to have low $P_T$), it is uncommon for $S$ to achieve high likelihood on them as they do not fit easily to the pattern learned by $S$. **See Supplementary for additional details and analysis.** Best viewed in color.

Additionally, we use the following methods/ models for evaluating the attack performance under SOTA MI defense methods: ● BiDO-HSIC [39][‡]. ● MID [9][§]. The details are included in Table 1.

**Evaluation Metrics.** Following [6, 11, 2], we use the following metrics to quantitatively evaluate the performance of MI attacks. Further, we also conduct user studies to assess the quality of reconstructed data (Sec. 6.5).

- *Attack Accuracy (Attack acc.)*: Following [1, 2, 6], we utilize an evaluation model, $E$, which employs a distinct architecture and is trained on $\mathcal{D}_{priv}$ [¶]. $E$ serves as a proxy for human inspection [1]. Higher attack accuracy indicates superior performance.

- *KNN Distance (KNN dt.)*: The KNN distance indicates the shortest distance between the reconstructed image of a specific identity and its private images. Specifically, the shortest distance is computed using the $l_2$ distance in the feature space, using the evaluation model's penultimate layer.

---

[‡] https://github.com/AlanPeng0897/Defend_MI

[§] https://github.com/Jiachen-T-Wang/mi-defense

[¶] Following previous work, $E$ can also be trained on $\mathcal{D}_{priv}$ and samples from additional identities. This could improve generalization of $E$ for accurate evaluation.

A smaller KNN distance signifies that the reconstructed images are more closely aligned with the private images of the target identity.

## 6.2 Training surrogate model with different algorithms

In this section, we demonstrate that our proposed T-ACGAN can lead to better surrogate models for MI. We describe a set of alternative approaches that can be used to train surrogate models using $\mathcal{D}_{pub}$ and compare the performance of these approaches with our proposed method. Specifically, we consider a set of five algorithms, which can be broadly classified into three categories, for learning the surrogate model $S$:

- **Directly use the public dataset** $\mathcal{D}_{pub}$. We present two methods to train $S$: • **Direct I.** We train $S$ using the public dataset labelled with target model, *i.e.* $\tilde{\mathcal{D}}_{pub}$ with samples $(x_p, \tilde{y})$, $x_p \in \mathcal{D}_{pub}$, $\tilde{y} = T(x_p)$; see Sec. 4.1. • **Direct II.** We apply data augmentation to $\tilde{\mathcal{D}}_{pub}$ of Direct I to reduce the class imbalance in Direct I, followed by training $S$ using the newly more balanced dataset.
- **Training an ACGAN.** We provide two versions: • **ACGAN I.** We train an ACGAN model on $\tilde{\mathcal{D}}_{pub}$ used in Direct I. • **ACGAN II.** We train an ACGAN model on augmented $\tilde{\mathcal{D}}_{pub}$ used in Direct II. As $C \circ D$ in ACGAN serves as a classifier, we use $C \circ D$ for MI attacks.
- **Training proposed T-ACGAN.** We use our proposed method described in Section 4.3 to train T-ACGAN. Similar to ACGAN I and II, we use $C \circ D$ after training T-ACGAN for the attack.

For this comparison, we utilize the following settings: $T =$ FaceNet64, $\mathcal{D}_{priv} =$ CelebA, $\mathcal{D}_{pub} =$ CelebA. Both ACGAN and T-ACGAN adopt the SNResnet architecture [34, 44]. To ensure a fair comparison, we use the same architecture as $C \circ D$ in ACGAN and T-ACGAN for the surrogate model $S$ in Direct I and Direct II. Detailed architecture specifications can be found in the Supplementary. After training the models, we employ the widely-used KEDMI [2] as the white-box attack on the trained surrogate models. Table 2 presents the results. The effectiveness of T-ACGAN in training surrogate models for MI attacks can be observed.

Table 2: We compare different approaches to train surrogate model for MI attacks. We utilize the following settings: $T =$ FaceNet64, $\mathcal{D}_{priv} =$ CelebA, $\mathcal{D}_{pub} =$ CelebA, and employ the KEDMI[2] for MI attacks.

| Algorithm | Attack acc. ↑ | KNN dt. ↓ |
|---|---|---|
| Direct I | $5.87 \pm 1.65$ | 1936.12 |
| Direct II | $9.60 \pm 2.22$ | 1890.16 |
| ACGAN I | $6.47 \pm 2.15$ | 1771.26 |
| ACGAN II | $7.87 \pm 3.10$ | 1785.20 |
| T-ACGAN | $42.07 \pm 3.46$ | 1473.99 |

## 6.3 Comparison against SOTA label-only MI attack

**Standard MI attack setup.** In this section, we present the results obtained from the standard attack setup on three datasets: CelebA, Facescrub, and Pubfig83, as detailed in Table 3. We evaluate three designs of surrogate: • (i) We directly use $C \circ D$ from our T-ACGAN as the surrogate model. The architecture of T-ACGAN can be found in the supplementary material. • (ii) We utilize the synthetic data generated by $G$ of our T-ACGAN and label it using the target classifier $T$ to train another surrogate model, denoted as $S =$ Densenet-161 [45]. • (iii) We employ the same data as in (ii) to train an ensemble of surrogate models, denoted as $S_{en}$, using different architectures including Densenet-121, Densenet-161, and Densenet-169.

We compare our results with the state-of-the-art (SOTA) label-only MI attack BREPMI [6]. To conduct our attacks, we utilize white-box PLGMI [7] on the surrogate models. Since PLGMI performs attacks using a conditional GAN trained with white-box access of the target classifier, we replace it with our T-ACGAN, which becomes available for use after training the surrogate models.

Our proposed method LOKT demonstrates a significant improvement in Attack accuracy and KNN distance compared to the SOTA label-only MI attack BREPMI [6]. Our top 1 attack accuracies are better than BREPMI from from 17.2% to 29.87% across all setups when we utilize the ensemble $S_{en}$.

Fig. 1 (f) presents a visual comparison of various methods under the setup $\mathcal{D}_{priv} =$ CelebA, $\mathcal{D}_{pub}$ = CelebA. **More results are available in the Supplementary**. Results clearly indicate that LOKT produces images that are closer to the ground truth (private data) compared to BREPMI [6]. This outcome provides strong evidence of the effectiveness of our approach in generating realistic images that closely resemble private data, which is critical for conducting successful MI attacks.

**MI attacks under large distribution shift.** Table 3 compares the MI attack results in the large distribution shift setup, where we use $\mathcal{D}_{pub} =$ FFHQ, $\mathcal{D}_{priv} =$ CelebA/ Facescrub/ Pubfig83, and

Table 3: We conduct comprehensive experiments to compare our proposed method LOKT and existing SOTA BREPMI [6] across standard MI attack benchmarks. Specifically, we evaluate the performance of our three proposed designs of surrogate, namely $C \circ D$, $S$, and $S_{en}$, while BREPMI performs black-box search on $T$ directly. We highlight the best results in each setup in **bold**.

| Setup | Attack | | Attack acc. ↑ | KNN dt. ↓ |
|---|---|---|---|---|
| $T$ = FaceNet64 $\mathcal{D}_{priv}$ = CelebA $\mathcal{D}_{pub}$ = CelebA | BREPMI | | 73.93 ± 4.98 | 1284.41 |
| | **LOKT** | $C \circ D$ | 81.00 ± 4.79 | 1298.63 |
| | | $S$ | 92.80 ± 2.59 | 1207.25 |
| | | $S_{en}$ | **93.93 ± 2.78** | **1181.72** |
| $T$ = IR152 $\mathcal{D}_{priv}$ = CelebA $\mathcal{D}_{pub}$ = CelebA | BREPMI | | 71.47 ± 5.32 | 1277.23 |
| | **LOKT** | $C \circ D$ | 72.07 ± 4.03 | 1358.94 |
| | | $S$ | 89.80 ± 2.33 | 1220.00 |
| | | $S_{en}$ | **92.13 ± 2.06** | **1206.78** |
| $T$ = VGG16 $\mathcal{D}_{priv}$ = CelebA $\mathcal{D}_{pub}$ = CelebA | BREPMI | | 57.40 ± 4.92 | 1376.94 |
| | **LOKT** | $C \circ D$ | 71.33 ± 4.39 | 1364.47 |
| | | $S$ | 85.60 ± 3.03 | 1252.09 |
| | | $S_{en}$ | **87.27 ± 1.97** | **1246.71** |
| $T$ = FaceNet64 $\mathcal{D}_{priv}$ = CelebA $\mathcal{D}_{pub}$ = FFHQ | BREPMI | | 43.00 ± 5.14 | 1470.55 |
| | **LOKT** | $C \circ D$ | 43.27 ± 3.53 | 1516.18 |
| | | $S$ | 59.13 ± 2.77 | 1437.86 |
| | | $S_{en}$ | **62.07 ± 3.89** | **1428.04** |

| Setup | Attack | | Attack acc. ↑ | KNN dt. ↓ |
|---|---|---|---|---|
| $T$ = FaceNet64 $\mathcal{D}_{priv}$ = Pubfig83 $\mathcal{D}_{pub}$ = Pubfig83 | BREPMI | | 55.60 ± 4.34 | 1012.83 |
| | **LOKT** | $C \circ D$ | 74.80 ± 5.93 | 924.58 |
| | | $S$ | 61.60 ± 3.58 | 993.44 |
| | | $S_{en}$ | **80.00 ± 3.16** | **883.52** |
| $T$ = FaceNet64 $\mathcal{D}_{priv}$ = Pubfig83 $\mathcal{D}_{pub}$ = FFHQ | BREPMI | | 72.80 ± 3.90 | 971.51 |
| | **LOKT** | $C \circ D$ | 85.60 ± 2.61 | 914.15 |
| | | $S$ | 88.40 ± 2.97 | 920.99 |
| | | $S_{en}$ | **94.40 ± 3.85** | **862.24** |
| $T$ = FaceNet64 $\mathcal{D}_{priv}$ = Facescrub $\mathcal{D}_{pub}$ = Facescrub | BREPMI | | 40.20 ± 6.60 | 1236.4 |
| | **LOKT** | $C \circ D$ | 45.70 ± 4.00 | 1296.29 |
| | | $S$ | 53.20 ± 5.29 | 1280.70 |
| | | $S_{en}$ | **58.60 ± 4.86** | **1225.13** |
| $T$ = FaceNet64 $\mathcal{D}_{priv}$ = Facescrub $\mathcal{D}_{pub}$ = FFHQ | BREPMI | | 37.30 ± 3.99 | 1456.59 |
| | **LOKT** | $C \circ D$ | 44.50 ± 5.98 | 1403.73 |
| | | $S$ | 47.20 ± 4.39 | 1404.85 |
| | | $S_{en}$ | **53.70 ± 4.57** | **1338.67** |

Table 4: We report Label-only MI Attack results under SOTA defense models namely BiDO [39] and MID [9]. We use $\mathcal{D}_{priv}$ = CelebA, $\mathcal{D}_{pub}$ = CelebA. We highlight the best results in **bold**.

| Setup | Attack | | Attack acc. ↑ | KNN dt. ↓ |
|---|---|---|---|---|
| $T$ = BiDO [39] $\mathcal{D}_{priv}$ = CelebA $\mathcal{D}_{pub}$ = CelebA | BREPMI[6] | | 37.40 ± 3.66 | 1500.45 |
| | **LOKT** | $C \circ D$ | 45.73 ± 5.94 | 1493.48 |
| | | $S$ | 58.53 ± 4.87 | 1427.22 |
| | | $S_{en}$ | **60.73 ± 3.07** | **1395.93** |
| $T$ = MID [9] $\mathcal{D}_{priv}$ = CelebA $\mathcal{D}_{pub}$ = CelebA | BREPMI[6] | | 39.20 ± 4.19 | 1458.61 |
| | **LOKT** | $C \circ D$ | 44.13 ± 3.54 | 1475.73 |
| | | $S$ | 55.33 ± 4.40 | 1393.76 |
| | | $S_{en}$ | **60.33 ± 4.76** | **1374.34** |

Table 5: The comparison of the number of queries (in millions) used by LOKT and BREPMI [6]. The attacks using $S$ and $S_{en}$ consume additional 500k queries comparing to $C \circ D$ to label the synthetic images to train $S$ and $S_{en}$. Our results show that we use fewer number of queries than BREPMI in all setups.

| $T$ | LOKT $C \circ D$ | LOKT $S/S_{en}$ | BREPMI |
|---|---|---|---|
| FaceNet64 | 12.16 | 12.66 | 17.98 |
| IR152 | 12.16 | 12.66 | 18.06 |
| VGG16 | 12.16 | 12.66 | 18.12 |
| BiDO-HSIC | 12.16 | 12.66 | 18.39 |
| MID | 12.16 | 12.66 | 18.25 |

$T$ = FaceNet64. The attack results of BREPMI drop significantly (by 30.93% for CelebA and 2.9% for Facescrub), while the results for Pubfig83 notably increase, which can be attributed to the small size of the Pubfig83 dataset [6]. Our proposed method outperforms BREPMI, with the top 1 attack accuracies increase from 16.40% to 21.60% for all setups. Moreover, the KNN distance indicates that our reconstructed images are closer to the private data than those reconstructed by BREPMI.

## 6.4 Additional results

**MI attack results using MI defense model.** We investigate the attacks on the MI defense model (see Table 4). Specifically, we utilize the SOTA defense model BiDO-HSIC [39] and MID [9]. Our results indicate that BiDO-HSIC successfully reduce the effectiveness of the white-box SOTA attack, PLGMI, by 9.57% (See the result in the Supplementary). In the label-only setup, the performance of BREPMI becomes relatively low with attack accuracy of only 37.40% for BiDO-HSIC [39] and 39.20% for MID [9]. In contrast, our approach achieves a much higher attack accuracy of 60.73% and 60.33%, almost doubling the performance of BREPMI. These results demonstrate that our approach is effective in conducting MI attacks on MI defense models, even in scenarios where the adversary has limited information about the target classifier.

**High resolution.** We conduct the experiment with high resolution images which has not been addressed yet for label-only setup [6]. In particular, we train a new target classifier $T$ = Resnet-152 using CelebA setup with the image size = 128×128. For fair comparison between BREPMI and our proposed method, T-ACGAN has the same GAN architectures used by BREPMI. The details of the architecture can be found in the Supplementary.

The results are shown in Table 6. LOKT outperforms BREPMI, with top 1 accuracy surpassing BREPMI by 20.27%. Our inverted images are closer to private training samples than BREPMI

Table 6: We conduct the experiment with higher resolution images. We use $T$ = Resnet-152, $\mathcal{D}_{priv}$ = CelebA, $\mathcal{D}_{pub}$ = CelebA, image size = 128×128. The natural accuracy of $T$ is 86.07%. We highlight the best results in **bold**.

| Setup | Attack | | Attack acc. ↑ | KNN dt. ↓ |
|---|---|---|---|---|
| $T$ = IR152 | BREPMI[6] | | $50.33 \pm 4.71$ | 1389.09 |
| $\mathcal{D}_{priv}$ = CelebA | | $C \circ D$ | $66.87 \pm 3.93$ | 1356.53 |
| $\mathcal{D}_{pub}$ = CelebA | **LOKT** | $S$ | $66.80 \pm 3.83$ | 1341.04 |
| | | $S_{en}$ | $\mathbf{70.60 \pm 4.43}$ | **1320.16** |

Table 7: User study results. Our human study reveals that users distinctly favor our approach, with 64.30% user preference for images reconstructed using our proposed approach, LOKT, compared to BREPMI's lower 35.70% user preference.

| Method | User Preference (↑) |
|---|---|
| BREPMI | 35.70% |
| LOKT | **64.30%** |

(smaller KNN distance). We believe our study can provide new insight on the effectiveness of SOTA label-only attack at a higher resolution of 128×128, paving the way to future label-only model inversion attacks at resolutions beyond 128×128.

**Query budget.** In this experiment, we compare query budget between our proposed method and BREPMI [2]. In the BREPMI, queries to the target classifier $T$ are required to identify the initial points for attacking and estimate the gradients during the attack. In our method, queries to $T$ are required to label the synthetic data during the training of T-ACGAN to obtain $C \circ D$, and additional 500k queries to label generated images of T-ACGAN to train $S$ and the ensemble $S_{en}$. For comparison, as shown in Table 5, we use $\mathcal{D}_{priv}$ = CelebA and $\mathcal{D}_{pub}$ = CelebA. The results show that our proposed method requires 30% fewer queries compared to BREPMI.

## 6.5 User study

**User study setup.** In this section, we go beyond objective metrics and consider subjective evaluation of MI attacks. In particular, we conduct a human study to understand the efficacy of our proposed method, LOKT, compared to BREPMI. We follow the setup by [10] for human study and use Amazon Mechanical Turk (MTurk) for experiments. The user interface is provided in the Supplementary. In this study, users are shown 5 real images of a person (identity) as reference. Then users are required to compare the 5 real images with two inverted images: one from our method (LOKT), the other from BREPMI. We use $D_{priv}$ = CelebA, $D_{pub}$ = CelebA and $T$ = FaceNet64. Following [10], we randomly selected 50 identities with 10 unique users evaluating each task accounting to 1000 comparison pairs.

**User study results.** We report the user study results in Table 7. Our human study reveals that users distinctly favor our approach, with 64.30% user preference for images reconstructed using our proposed approach, in contrast to BREPMI's lower 35.70% user preference. These subjective evaluations further show the efficacy of our proposed method, LOKT, in the challenging label-only MI setup.

# 7 Discussion

**Conclusion.** Instead of performing a black-box search approach as in existing SOTA, we propose a new label-only MI approach (LOKT) by transferring decision knowledge from the target model to surrogate models and performing white-box attacks on the surrogate models. To obtain the surrogate models, we propose a new T-ACGAN to leverage generative modeling and the target model for effective knowledge transfer. Using findings of general learning dynamics of DNNs, we conduct analysis to support that our surrogate models are effective proxies for the target model under MI. We perform extensive experiments and ablation to support our claims and demonstrate significant improvement over existing SOTA.

**Broader Impacts.** Understanding model inversion attacks holds significance as AI models continue to see widespread deployment across various applications. By studying and understanding the approaches and methodologies for model inversion, researchers can develop good practices in deploying AI models and robust defense mechanisms for different applications esp. those involving sensitive training data. It is important to emphasize that the objective of model inversion research is to raise awareness of potential privacy threats and bolster our collective defenses.

**Limitations.** While our experiments are extensive compared to previous works, practical applications involve different types of private training datasets such as healthcare data. Nevertheless, our assumptions are general, and we believe our findings can be applied to a broader range of applications.

**Acknowledgements.** This research is supported by the National Research Foundation, Singapore under its AI Singapore Programmes (AISG Award No.: AISG2-TC-2022-007) and SUTD project PIE-SGP-AI-2018-01. This research work is also supported by the Agency for Science, Technology and Research (A*STAR) under its MTC Programmatic Funds (Grant No. M23L7b0021). This material is based on the research/work support in part by the Changi General Hospital and Singapore University of Technology and Design, under the HealthTech Innovation Fund (HTIF Award No. CGH-SUTD-2021-004). We thank anonymous reviewers for their insightful feedback.

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
