# Label-Only Model Inversion Attacks via Knowledge Transfer Supplementary Materials

**Ngoc-Bao Nguyen**[*1]     **Keshigeyan Chandrasegaran**[*2‡]
**Milad Abdollahzadeh**[1]     **Ngai-Man Cheung**[1†]
[1]Singapore University of Technology and Design (SUTD)     [2]Stanford University
thibaongoc_nguyen@mymail.sutd.edu.sg  ngaiman_cheung@sutd.edu.sg

In this supplementary material, we provide additional experiments, analysis, ablation study, and details required to reproduce our results. Pytorch code, demo, pre-trained models and reconstructed data are available at our project website.

## Contents

---

[*] These authors contributed equally.     [‡] Work done while at SUTD.
[†] Corresponding author.

37th Conference on Neural Information Processing Systems (NeurIPS 2023).

# A    Additional Analysis and Visualizations

## A.1    Our Surrogate models are effective proxies for the opaque Target model for MI

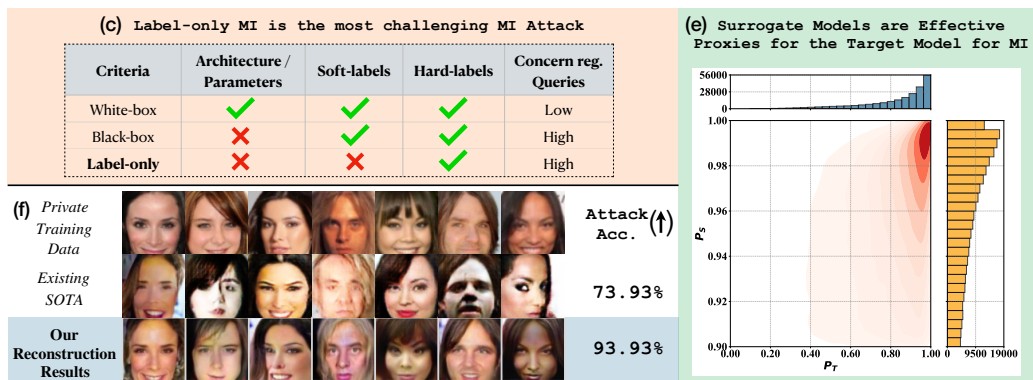

Figure A.1: We use $\mathcal{D}_{priv}$ = CelebA, $\mathcal{D}_{pub}$ = CelebA, $T$ = FaceNet64, $S$ = DenseNet-161. **(c)** We cast the challenging problem setup of label-only MI attack as a white-box MI attack. To our knowledge, our proposed approach is the first to address label-only MI via white-box MI attacks. **(e)** We consider high likelihood samples under $S$. i.e.: $P_S > 0.9$. Our analysis using 500k training data demonstrates that $S$ is an effective proxy for $T$ for MI attack. In particular, the white-box MI attack on $S$ mimics the white-box attack on opaque $T$. **(f)** Additional reconstruction results using our proposed approach ($S_{en}$). We remark that our proposed approach significantly improves the Label-only MI attack (e.g. $\approx 20\%$ improvement in standard CelebA benchmark compared to existing SOTA [1]) resulting in significant improvement in private data reconstruction. Best viewed in color.

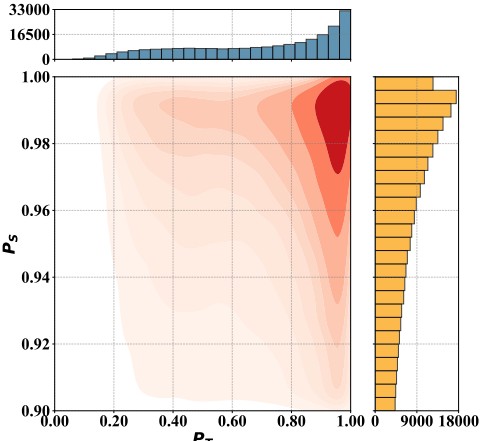

Figure A.2: Figure 1 (e) from the main paper supports that $S$ is a good proxy for $T$ for MI established using **Property P1**. We use $\mathcal{D}_{priv}$ = CelebA, $\mathcal{D}_{pub}$ = CelebA, $T$ = FaceNet64, $S$ = DenseNet-161. We use 500k validation data for analysis.

**White-box MI attack on S mimics the white-box attack on T.** For clarity, we copy Figure 1(e) (main paper) to Supplementary Fig. A.2. In this section, we include the details of Fig. A.2 and provide additional empirical evidence in Figure A.1(e) to support **Property P1**. We remark that Fig. A.2 and Fig. A.1(e) use 500k validation and 500k training data respectively[*]. In both figures, we consider high-likelihood samples under $S$. i.e.: $P_S > 0.90$. We remark that since in our framework, we optimize white-box attack w.r.t. $S$, the reconstructed samples usually have a very high likelihood under $S$ (above 0.9). Therefore, we condition our analysis on $P_S > 0.9$. As one can clearly observe in both conditional $P_T$ histograms in Fig. A.2 and Fig. A.1(e), high likelihood samples under $S$ are

---

[*]We recall that the data samples are generated samples from our T-ACGAN. Using generated samples for analysis is suitable as generated samples are utilized during white-box MI.

likely to have high likelihood under $T$ **(Property P1)**, and it is uncommon for high likelihood samples under $S$ to have low likelihood under $T$. Given **P1**, white-box attacks on $S$ can mimic white-box attacks on $T$, resulting in $S$ being an effective proxy for $T$ for MI. In addition, we report similar observations on another setup: $\mathcal{D}_{priv}$=CelebA, $\mathcal{D}_{pub}$=FFHQ, $T$=FaceNet64, $S$=DenseNet-161 in Fig. A.3.

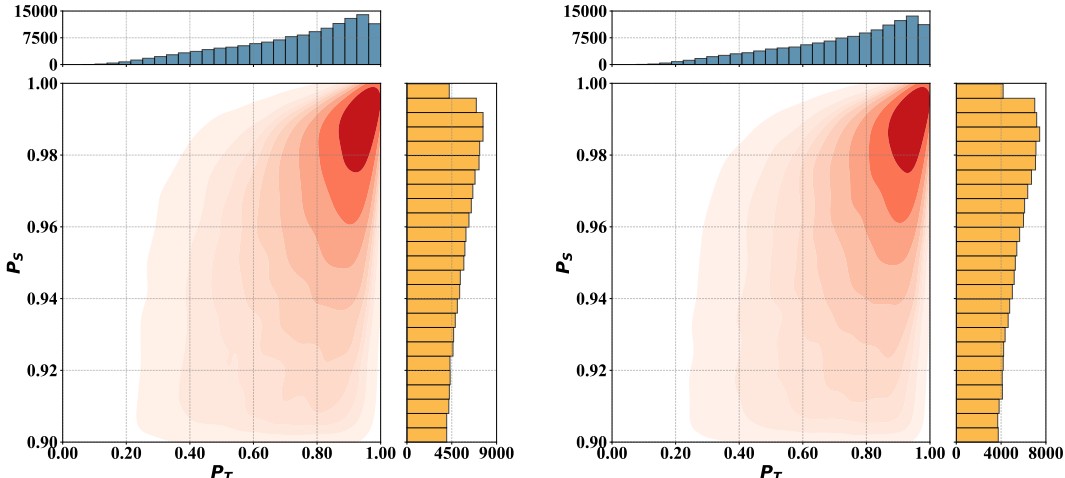

Figure A.3: We use $\mathcal{D}_{priv}$ = CelebA, $\mathcal{D}_{pub}$ = FFHQ, $T$ = FaceNet64, $S$ = DenseNet-161. we consider high likelihood samples under $S$. i.e.: $P_S > 0.90$, and show results for 500k training samples (left) and 500k validation samples (right). As one can clearly observe in both conditional $P_T$ histograms, high likelihood samples under $S$ are likely to have high likelihood under $T$ **(Property P1)**, and it is uncommon for high likelihood samples under $S$ to have low likelihood under $T$. Given **P1**, white-box attacks on $S$ can mimic white-box attacks on $T$, resulting in $S$ being a an effective proxy of $T$ for MI.

**Why would $S$ possess P1?** We provide additional empirical results using training and validation sets to support why $S$ possesses **P1** using the framework by [2]. We use publicly available SOTA face recognition model(s) [†] to extract face embeddings (128-dimensional) for analysis. We use the following setup for analysis: $\mathcal{D}_{priv}$ = CelebA [3], $\mathcal{D}_{pub}$ = CelebA [3], $T$ = FaceNet64, $S$ = DenseNet-161. Based on the distance from the face-embedding centroid for each identity, we consider the closest 70% of samples as easy samples, and the remaining 30% samples as hard samples [‡]. The training dynamic results for easy and hard samples for 3 sets of randomly chosen identities are shown in Fig. A.4, A.5 and A.6, for both training and validation sets. We also show the training dynamics for the validation set corresponding to the main paper analysis results in Fig. A.7.

## A.2 Decision knowledge transfer to T-ACGAN during training

In this section, we provide additional analysis to support that the target model, $T$'s, decision knowledge is adequately transferred to our T-ACGAN during training. Following the definition in Sec. 4.3 (main paper), $x_f = G(z, y)$, $\tilde{y} = T(x_f)$, let $\gamma$ be the percentage of samples with $y$ the same as $\tilde{y}$ in a batch of samples. In particular, we track $\gamma$ throughout our T-ACGAN training. Initially, we expect $\gamma$ to be low and with increasing training iterations, we expect $\gamma$ to increase indicating adequate decision knowledge transfer from the target model $T$. We report $\gamma$ tracking results for 2 experiment setups in Fig. A.8. ● **Setup 1:** We use $\mathcal{D}_{priv}$ = CelebA [3], $\mathcal{D}_{pub}$ = CelebA [3], $T$ = FaceNet64. ● **Setup 2:** We use $\mathcal{D}_{priv}$ = CelebA [3], $\mathcal{D}_{pub}$ = FFHQ [4], $T$ = FaceNet64. We remark that batch size=128 as we track $\gamma$ for every batch. We train T-ACGAN for 100k iterations. As one can observe, $\gamma$ starts low and gradually increases during T-ACGAN training indicating adequate knowledge transfer from $T$.

---

[†] https://github.com/ageitgey/face_recognition

[‡] Note that the 70:30 selection of easy:hard samples has no effect to our algorithm; in fact our algorithm does not need explicit separation of easy/hard samples. Here in this discussion, we separate easy and hard samples only to ease our illustration of *different pace of $P_S$ improvement among the samples*, which results in most samples with $P_S > 0.9$ having high $P_T$.

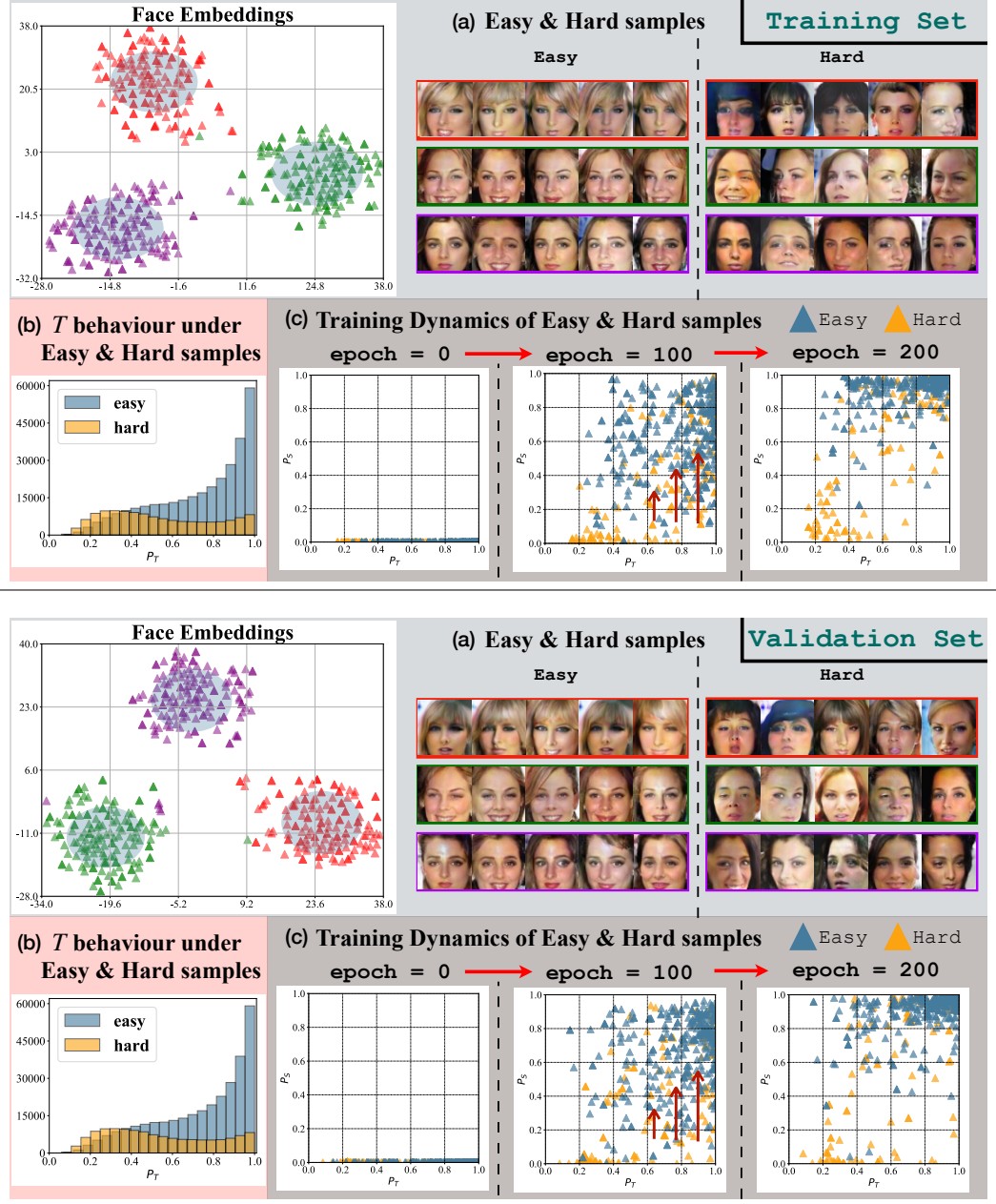

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

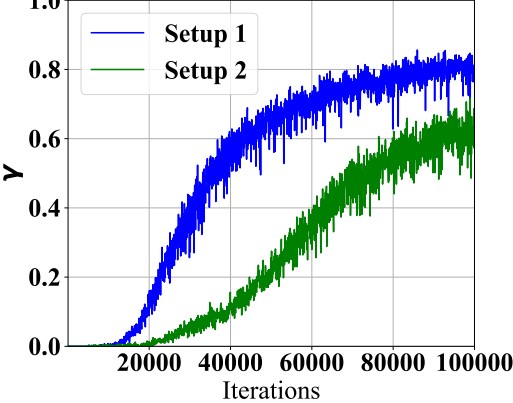

Figure A.8: We report $\gamma$ tracking results during T-ACGAN training for 2 setups. Our T-ACGAN is trained for 100k iterations. In both setups, ● $\gamma$ starts low ($\gamma \approx 0$ for iterations less than 5000) ● With increasing iterations, $\gamma$ increases indicating adequate decision knowledge transfer from the target model $T$ to T-ACGAN. We remark that **Setup 2** has lower $\gamma$ in general compared to **Setup 1** due to a large distribution shift between public data and private data.

# B Additional results

## B.1 Different white-box attacks with surrogate models

In this section, we perform a set of experiments to demonstrate that the surrogate models trained using our proposed framework are versatile enough to be used with different white-box MI attacks. For this analysis, we use two SOTA white-box attacks, namely KEDMI [5] and PLGMI [6]. For each white-box attack, we train five different surrogate models using our proposed framework including: $C \circ D$, $S_{DN121}$ = Desenet-121, $S_{DN161}$ = Desenet-161, $S_{DN169}$ = Desenet-169, and $S_{en}$ = {Desenet-121, Desenet-161, Desenet-169}, and then, evaluate the white-box MI attack performance on each of these surrogate models.

In the case of KEDMI, we train a specific-GAN using our surrogate model $S$ using the official implementation [§]. As for PLGMI [¶], given that our T-ACGAN can serve as a replacement for the conditional GAN of PLGMI, we leverage our T-ACGAN to apply the PLGMI attack. It is noteworthy that the target classifier $T$ is not used during the attacks when we apply white-box attacks on our surrogate models.

We report the results in Table B.1, utilizing the CelebA dataset setup. Our results demonstrate the effectiveness of our surrogate models using white-box MI attacks, and are consistent with the outcomes obtained using the target classifier $T$ in white-box attacks.

Table B.1: We compare the attack results using different white-box attacks with five surrogate models. We use $T$ = FaceNet64, $\mathcal{D}_{priv}$ = CelebA, $\mathcal{D}_{pub}$ = CelebA. The results show that our different designs of surrogate model perform well across different white-box attacks. Note that the white-box attack results on $T$ are included only as reference as our setup does not have access to $T$ parameters nor soft-label of $T$.

| Attack | Model | Attack acc. ↑ | KNN dt. ↓ |
|--------|-------|---------------|-----------|
| KEDMI | $T$ [5] | $81.13 \pm 4.66$ | 1298.63 |
| | $C \circ D$ | $42.07 \pm 3.46$ | 1473.99 |
| | $S_{DN121}$ | $62.93 \pm 4.67$ | 1350.67 |
| | $S_{DN161}$ | $65.07 \pm 3.79$ | 1351.07 |
| | $S_{DN169}$ | $62.80 \pm 4.45$ | 1350.56 |
| | $S_{en}$ | $69.00 \pm 4.03$ | 1329.84 |
| PLGMI | $T$ [6] | $99.00 \pm 0.01$ | 1103.03 |
| | $C \circ D$ | $81.00 \pm 4.79$ | 1298.63 |
| | $S_{DN121}$ | $92.27 \pm 2.85$ | 1208.55 |
| | $S_{DN161}$ | $92.80 \pm 2.59$ | 1207.25 |
| | $S_{DN169}$ | $92.33 \pm 3.36$ | 1206.15 |
| | $S_{en}$ | $93.93 \pm 2.78$ | 1181.72 |

## B.2 Different TACGAN architecture

For a fair comparison with BREPMI [1], we provide the experiment results by training a new T-ACGAN using the same architectures as the GAN used by BREPMI. For the discriminator (D), we apply max pooling and add a linear layer before the last layer for the classifier head. As for the generator (G), we retain the same architecture and replace batch normalization with conditional batch normalization.

We report the results in Table B.2. Our results are better than BREPMI when using the same GAN architecture.

## B.3 White-box attack results for reference

We show the our proposed method and other SOTA white-box attacks including GMI [7], KEDMI [5], PLGMI [6], and the SOTA label-only attack BREPMI [1] in Table B.3 for reference.

---

[§]https://github.com/SCccc21/Knowledge-Enriched-DMI
[¶]https://github.com/LetheSec/PLG-MI-Attack

Table B.2: We conduct comprehensive comparison between our proposed method and existing SOTA BREPMI [1] using the same GAN architecture. Specifically, we evaluate the performance of our three proposed designs of surrogate, namely $C \circ D$, $S$, and $S_{en}$, while BREPMI performs black-box search on $T$ directly. We highlight the best results in each setup in **bold**.

| Setup | | Attack | | Attack acc. ↑ | KNN dt. ↓ |
|---|---|---|---|---|---|
| $T$ = FaceNet64 $\mathcal{D}_{priv}$ = CelebA $\mathcal{D}_{pub}$ = CelebA | | BREPMI | | $73.93 \pm 4.98$ | 1284.41 |
| | | | $C \circ D$ | $85.47 \pm 2.95$ | 1336.45 |
| | | **LOKT** | $S$ | $90.73 \pm 3.57$ | 1251.16 |
| | | | $S_{en}$ | $\mathbf{93.20 \pm 1.98}$ | **1214.60** |
| $T$ = IR152 $\mathcal{D}_{priv}$ = CelebA $\mathcal{D}_{pub}$ = CelebA | | BREPMI | | $71.47 \pm 5.32$ | 1277.23 |
| | | | $C \circ D$ | $88.20 \pm 3.48$ | 1304.05 |
| | | **LOKT** | $S$ | $92.27 \pm 2.46$ | 1236.87 |
| | | | $S_{en}$ | $\mathbf{94.53 \pm 2.34}$ | **1214.38** |
| $T$ = VGG16 $\mathcal{D}_{priv}$ = CelebA $\mathcal{D}_{pub}$ = CelebA | | BREPMI | | $57.40 \pm 4.92$ | 1376.94 |
| | | | $C \circ D$ | $68.93 \pm 4.23$ | 1450.74 |
| | | **LOKT** | $S$ | $78.07 \pm 2.91$ | 1362.70 |
| | | | $S_{en}$ | $\mathbf{82.80 \pm 3.20}$ | **1346.51** |
| $T$ = FaceNet64 $\mathcal{D}_{priv}$ = CelebA $\mathcal{D}_{pub}$ = FFHQ | | BREPMI | | $43.00 \pm 5.14$ | 1470.55 |
| | | | $C \circ D$ | $59.87 \; 5.05$ | 1509.09 |
| | | **LOKT** | $S$ | $67.20 \pm 4.23$ | 1467.62 |
| | | | $S_{en}$ | $\mathbf{72.33 \pm 3.30}$ | **1454.43** |

## B.4    Model stealing

One related area to training surrogate models for a target model is *model stealing* where an attacker aims to copy the performance of a target model. In this section, we compare the performance of our proposed method for training surrogate models —specifically designed for MI attacks— with model stealing approaches. More specifically, we apply the SOTA model stealing approach DFMS-HL [‖] [9] that only uses the hard labels to train the surrogate model $S$. We train two surrogate models $S = C \circ D$, and $S =$ Densenet-161 [10] using DFMS-HL and compare it with the trained surrogate models with the proposed approach. Table B.4 shows that the surrogate models trained with our proposed method can perform much better for MI attacks.

## B.5    Architecture selection for Surrogate models

Our proposed approach (casting label-only MI attack as white-box MI attack) allows the possibility for MI attackers to choose the surrogate model architecture(s). In this section, we study the effect of model architectures on model accuracy and MI attack accuracy to empirically justify our use of DenseNet model variants [10] as surrogate models. The details of this study is as follows: We conduct MI attacks on three different model families including MobileNet (MobileNetV2 [11] and MobileNetV3-small/ large [12]), EfficientNet [13] (EfficientNet-B0, EfficientNet-B1, EfficientNet-B2, EfficientNet-B3, EfficientNet-B4, EfficientNet-B7), and DenseNet [10] (DenseNet-121, DenseNet-161, DenseNet-169). The number of parameters for each model (in Millions) is given in Table B.5. We first train these 12 model architectures using private dataset $\mathcal{D}_{priv}$ = CelebA [3] which contains 30,027 images/ 1,000 identities following the exact training protocol in [5].

After training target models, we perform white-box MI attacks on these target models. We use two popular white-box MI attacks namely GMI [7] and KEDMI [5]. Following [5], we use evaluation model $E =$ FaceNet [14]. We report the model accuracy and MI attack accuracy in Fig. B.1. When comparing models within the same family, in general, we observe that architectures with more parameters achieve better model accuracy and are more susceptible to MI attacks (Higher MI Attack Acc). Based on KEDMI [5] results obtained in this study, *we use architectures from the DenseNet family*[**].

---

[‖]https://github.com/val-iisc/Hard-Label-Model-Stealing
[**]DenseNet-161 has more parameters than DenseNet-169 More details: as our surrogate model(s)

Table B.3: We evaluate the performance of our label-only attack method across various experimental setups. For reference, we also include our results against three state-of-the-art (SOTA) white-box attacks, namely GMI [7], KEDMI [5], PLGMI [6], as well as the SOTA label-only attack BREPMI [1]. The obtained results clearly demonstrate the effectiveness of our label-only attack method over BREPMI, while also achieving comparable performance with other white-box attacks.

| | Label-only MI Attacks | | | | White-box MI Attacks (for reference only) | | | | | |
| | LOKT | | BREPMI [1] | | GMI [7] | | KEDMI [5] | | PLGMI [6] | |
| S | Attack acc. ↑ | KNN dt. ↓ | Attack acc. ↑ | KNN dt. ↓ | Attack acc. ↑ | KNN dt. ↓ | Attack acc. ↑ | KNN dt. ↓ | Attack acc. ↑ | KNN dt. ↓ |
|---|---|---|---|---|---|---|---|---|---|---|
| $T$ = **FaceNet64**, $\mathcal{D}_{priv}$ = **CelebA**, $\mathcal{D}_{pub}$ = **CelebA** | | | | | | | | | | |
| $C \circ D$ | $81.00 \pm 4.79$ | 1298.63 | | | | | | | | |
| $S$ | $92.80 \pm 2.59$ | 1207.25 | $73.93 \pm 4.98$ | 1284.41 | $26.20 \pm 4.66$ | 1626.60 | $81.13 \pm 4.66$ | 1247.91 | $99.00 \pm 0.01$ | 1103.03 |
| $S_{en}$ | $93.93 \pm 2.78$ | 1181.72 | | | | | | | | |
| $T$ = **IR152**, $\mathcal{D}_{priv}$ = **CelebA**, $\mathcal{D}_{pub}$ = **CelebA** | | | | | | | | | | |
| $C \circ D$ | $72.07 \pm 4.03$ | 1358.94 | | | | | | | | |
| $S$ | $89.80 \pm 2.33$ | 1220.00 | $71.47 \pm 5.32$ | 1277.23 | $29.47 \pm 4.70$ | 1609.57 | $79.87 \pm 3.52$ | 1251.37 | $100.0 \pm 0.00$ | 1026.71 |
| $S_{en}$ | $92.13 \pm 2.06$ | 1206.78 | | | | | | | | |
| $T$ = **VGG16**, $\mathcal{D}_{priv}$ = **CelebA**, $\mathcal{D}_{pub}$ = **CelebA** | | | | | | | | | | |
| $C \circ D$ | $71.33 \pm 4.39$ | 1364.47 | | | | | | | | |
| $S$ | $85.60 \pm 3.03$ | 1252.09 | $57.40 \pm 4.92$ | 1376.94 | $18.07 \pm 4.44$ | 1705.04 | $74.07 \pm 4.21$ | 1290.81 | $97.00 \pm 0.01$ | 1120.61 |
| $S_{en}$ | $87.27 \pm 1.97$ | 1246.71 | | | | | | | | |
| $T$ = **BiDO-HSIC [8]**, $\mathcal{D}_{priv}$ = **CelebA**, $\mathcal{D}_{pub}$ = **CelebA** | | | | | | | | | | |
| $C \circ D$ | $45.73 \pm 5.94$ | 1493.48 | | | | | | | | |
| $S$ | $58.53 \pm 4.87$ | 1427.22 | $37.40 \pm 3.66$ | 1500.45 | $5.93 \pm 1.85$ | 1930.52 | $42.80 \pm 4.58$ | 1478.32 | $87.53 \pm 3.08$ | 1237.41 |
| $S_{en}$ | $60.73 \pm 3.07$ | 1395.93 | | | | | | | | |
| $T$ = **FaceNet64**, $\mathcal{D}_{priv}$ = **Facescrub**, $\mathcal{D}_{pub}$ = **Facescrub** | | | | | | | | | | |
| $C \circ D$ | $45.70 \pm 4.00$ | 1296.29 | | | | | | | | |
| $S$ | $53.20 \pm 5.29$ | 1280.70 | $40.20 \pm 6.60$ | 1236.40 | $14.60 \pm 3.70$ | 1599.67 | $55.20 \pm 4.61$ | 1193.41 | $92.50 \pm 2.91$ | 1012.74 |
| $S_{en}$ | $58.60 \pm 4.86$ | 1225.13 | | | | | | | | |
| $T$ = **FaceNet64**, $\mathcal{D}_{priv}$ = **Pubfig83**, $\mathcal{D}_{pub}$ = **Pubfig83** | | | | | | | | | | |
| $C \circ D$ | $74.80 \pm 5.93$ | 924.58 | | | | | | | | |
| $S$ | $61.60 \pm 3.58$ | 995.08 | $55.60 \pm 4.34$ | 1012.83 | $16.40 \pm 4.77$ | 1338.61 | $66.00 \pm 4.00$ | 1031.86 | $99.60 \pm 0.89$ | 832.07 |
| $S_{en}$ | $80.00 \pm 3.16$ | 883.52 | | | | | | | | |
| $T$ = **FaceNet64**, $\mathcal{D}_{priv}$ = **CelebA**, $\mathcal{D}_{pub}$ = **FFHQ** | | | | | | | | | | |
| $C \circ D$ | $43.27 \pm 3.53$ | 1516.18 | | | | | | | | |
| $S$ | $59.13 \pm 2.77$ | 1437.86 | $43.00 \pm 5.14$ | 1470.55 | $11.00 \pm 4.64$ | 1750.74 | $54.20 \pm 5.16$ | 1443.44 | $95.00 \pm 0.04$ | 1241.41 |
| $S_{en}$ | $62.07 \pm 3.89$ | 1428.04 | | | | | | | | |
| $T$ = **FaceNet64**, $\mathcal{D}_{priv}$ = **Facescrub**, $\mathcal{D}_{pub}$ = **FFHQ** | | | | | | | | | | |
| $C \circ D$ | $44.50 \pm 5.98$ | 1403.73 | | | | | | | | |
| $S$ | $47.20 \pm 4.39$ | 1404.85 | $37.30 \pm 3.99$ | 1456.59 | $11.00 \pm 3.63$ | 1864.71 | $50.80 \pm 4.58$ | 1337.96 | $89.10 \pm 3.05$ | 1196.88 |
| $S_{en}$ | $53.70 \pm 4.57$ | 1338.67 | | | | | | | | |
| $T$ = **FaceNet64**, $\mathcal{D}_{priv}$ = **Pubfig83**, $\mathcal{D}_{pub}$ = **FFHQ** | | | | | | | | | | |
| $C \circ D$ | $85.60 \pm 2.61$ | 914.15 | | | | | | | | |
| $S$ | $88.40 \pm 2.97$ | 920.99 | $72.80 \pm 3.90$ | 971.51 | $36.40 \pm 5.55$ | 1199.00 | $84.00 \pm 4.00$ | 891.21 | $100.0 \pm 0.00$ | 787.57 |
| $S_{en}$ | $94.40 \pm 3.85$ | 862.24 | | | | | | | | |

Table B.4: The comparison on MI attacks results using our surrogate models and model stealing DFMS-HL [9]. Here we use PLGMI [6], $\mathcal{D}_{priv}$ = CelebA, $\mathcal{D}_{pub}$ = CelebA, $T$ = FaceNet64.

| S | DFMS-HL [9] | | LOKT | |
| | Attack Acc. ↑ | KNN dt. ↓ | Attack Acc. ↑ | KNN dt. ↓ |
|---|---|---|---|---|
| $C \circ D$ | $14.00 \pm 4.01$ | 1775.71 | $81.00 \pm 4.79$ | 1298.63 |
| Densenet-161 | $67.13 \pm 3.67$ | 1411.45 | $92.80 \pm 2.59$ | 1207.25 |

Table B.5: Number of parameters (in Millions) for different model architectures.

| Model | Mobi_small | Mobi_large | Mobi_v2 | Eff-B0 | Eff-B1 | Eff-B2 | Eff-B3 | Eff-B4 | Eff-B7 | Den-121 | Den-161 | Den-169 |
|---|---|---|---|---|---|---|---|---|---|---|---|---|
| Parameters (M) | 1.50 | 3.93 | 3.50 | 5.29 | 7.79 | 9.11 | 12.20 | 19.30 | 66.30 | 11.10 | 35.30 | 19.10 |

# C    Additional Reconstruction Results

In this section, we show reconstructed samples for 3 additional setups using our proposed method. We show cross-dataset MI results in Fig. C.2 using FaceNet64 target model. In addition, we also

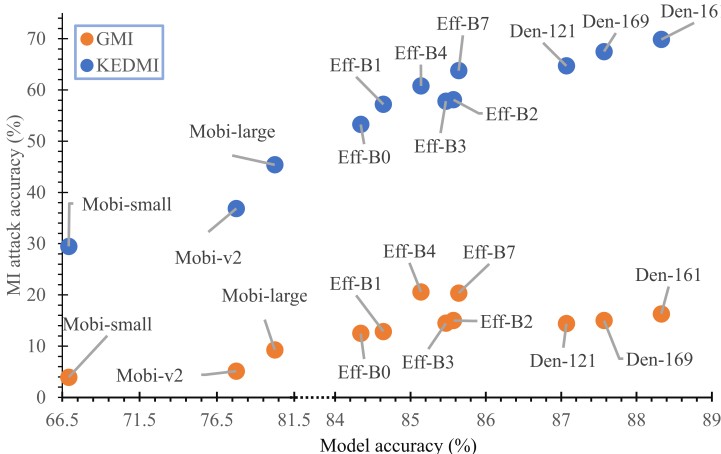

Figure B.1: *Architecture selection for surrogate models:* We report model accuracy and MI attack accuracy of 12 models from 3 model families namely, MobileNet (MobileNetV2 [11] and MobileNetV3-small/ large [12]), EfficientNet [13] (EfficientNet-B0, EfficientNet-B1, EfficientNet-B2, EfficientNet-B3, EfficientNet-B4, EfficientNet-B7), and DenseNet [10] (DenseNet-121,DenseNet-161, DenseNet-169). The number of parameters for each model is included in Table B.5. ● We observe that compact models such as MobileNets obtain relatively lower model accuracy and lower MI Attack accuracy. ● We observe that larger models, i.e.: DenseNet models, achieve relatively higher model accuracy and higher MI Attack accuracy making them good candidates for surrogate models.

show results for 2 additional target models: IR152 [15] and VGG16 [16] in Fig. C.3 and Fig. C.4 respectively.

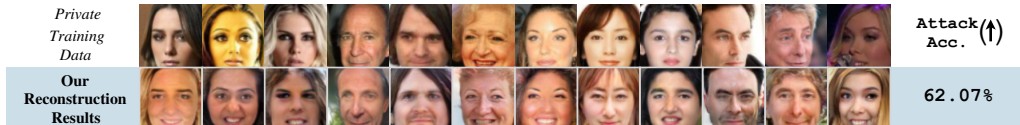

Figure C.2: We use $\mathcal{D}_{priv}$ = CelebA [3], $\mathcal{D}_{pub}$ = FFHQ [4], $T$ = FaceNet64. We show private data (top), *our* reconstruction results (bottom) and Attack accuracy. We remark that these results are obtained using $S_{en}$.

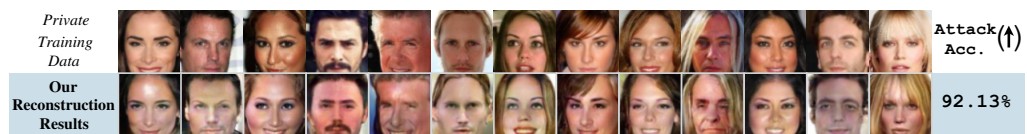

Figure C.3: We use $\mathcal{D}_{priv}$ = CelebA [3], $\mathcal{D}_{pub}$ = CelebA [3], $T$ = IR152 [15]. We show private data (top), *our* reconstruction results (bottom) and Attack accuracy. We remark that these results are obtained using $S_{en}$.

# D   Experiment details/ Design choices

We use three datasets including CelebA [3], Facescrub [17], and Pubfig83 [18]. We further examine the distribution shift by using FFHQ dataset [4] which includes images that vary in terms of background, ethnicity, and age. Following [5, 1], we divide CelebA into two datasets $\mathcal{D}_{priv}$ for training the target model $\mathcal{T}$ and $\mathcal{D}_{pub}$ for training GAN and surrogate models $\mathcal{C}$. The details of each dataset are summarized in Table D.6.

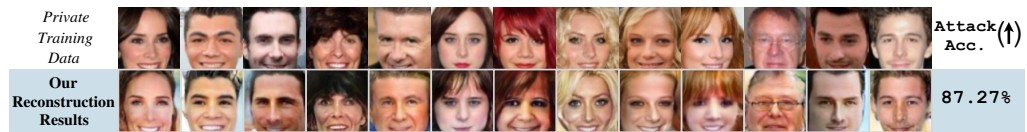

Figure C.4: We use $\mathcal{D}_{priv}$ = CelebA [3], $\mathcal{D}_{pub}$ = CelebA [3], $T$ = VGG16 [16]. We show private data (top), *our* reconstruction results (bottom) and Attack accuracy. We remark that these results are obtained using $S_{en}$.

Table D.6: Details of three datasets including CelebA [3], Facescrub [17], and Pubfig83 [18].

| Dataset | $\mathcal{D}_{priv}$ | | $\mathcal{D}_{pub}$ | |
|---|---|---|---|---|
| | # Target id | # Images | # Id | # Images |
| CelebA [3] | 1,000 | 30,027 | - | 30,000 |
| Facescrub [17] | 200 | 40,953 | 330 | 65,910 |
| Pubfig83 [18] | 50 | 8,145 | 33 | 5,693 |
| FFHQ [4] | - | - | - | 70,000 |

# E   Evaluation details

Following [1], we attack the first 300 out of 1000 labels in the experiments using CelebA dataset. In cases of Facescrub and Pubfig, we attack all the labels of the target classifier (200 and 50, respectively). As for the evaluation model, we use FaceNet which is trained on the private dataset and has higher resolution than the target classifier (image resolution 112x112). We remark that all pre-trained target/ evaluation models are released publicly by [1], and we adopt these models in our experiments for fair comparison.

## E.1   T-ACGAN architecture

We adopt the SNResnet architecture [19, 20] for our T-ACGAN. The architecture of the generator and the discriminator are as shown in Table E.7 and E.8.

Table E.7: Generator

| Operation | Kernel | Strides | Feature maps | BN? |
|---|---|---|---|---|
| Linear | N/A | N/A | 16384 | |
| Convolution | 3x3 | 1x1 | 512 | |
| Convolution | 3x3 | 1x1 | 512 | yes |
| Convolution | 1x1 | 1x1 | 512 | |
| Convolution | 3x3 | 1x1 | 256 | |
| Convolution | 3x3 | 1x1 | 256 | yes |
| Convolution | 1x1 | 1x1 | 256 | |
| Convolution | 3x3 | 1x1 | 128 | |
| Convolution | 3x3 | 1x1 | 128 | yes |
| Convolution | 1x1 | 1x1 | 128 | |
| Convolution | 3x3 | 1x1 | 64 | |
| Convolution | 3x3 | 1x1 | 64 | yes |
| Convolution | 1x1 | 1x1 | 64 | yes |
| Convolution | 1x1 | 1x1 | 3 | |

## E.2   Hyperparameters

**Training T-ACGAN.** The T-ACGAN model was trained using different numbers of iterations for CelebA [3], Facescrub [17], and Pubfig83 [18] datasets. Specifically, we utilized 20k iterations for CelebA, 5k iterations for Facescrub, and 3k iterations for Pubfig83. It's important to note that during training, the generator $G$ was trained once while the discriminator $D$ was trained five times for each iteration. For T-ACGAN loss, including generator loss $\mathcal{L}_G$, and discriminator loss $\mathcal{L}_{D,C}$

Table E.8: Discriminator. $N$ is the number of classes.

| Operation | Kernel | Strides | Feature maps |
|-----------|--------|---------|--------------|
| Convolution | 3x3 | 1x1 | 64 |
| Convolution | 3x3 | 1x1 | 64 |
| Convolution | 1x1 | 1x1 | 64 |
| Convolution | 3x3 | 1x1 | 64 |
| Convolution | 3x3 | 1x1 | 128 |
| Convolution | 1x1 | 1x1 | 128 |
| Convolution | 3x3 | 1x1 | 128 |
| Convolution | 3x3 | 1x1 | 256 |
| Convolution | 1x1 | 1x1 | 256 |
| Convolution | 3x3 | 1x1 | 256 |
| Convolution | 3x3 | 1x1 | 512 |
| Convolution | 1x1 | 1x1 | 512 |
| Convolution | 3x3 | 1x1 | 512 |
| Convolution | 3x3 | 1x1 | 1024 |
| Convolution | 1x1 | 1 | 1024 |
| Linear | N/A | N/A | 1 |
| Linear | N/A | N/A | N |

(Eqn. (3) and (4) in the main paper), we select $\lambda_1 = 1.0$ and $\lambda_2 = 1.5$ for all experiments. This deliberate choice aims to enhance the learning process of both the generator and the discriminator by emphasizing the importance of conditional loss.

$$\mathcal{L}_G = \lambda_1 E[\log P(s = Fake|x_f)] - \lambda_2 E[\log P(c = y|x_f)]$$

$$\mathcal{L}_{D,C} = -\lambda_1[E[\log P(s = Fake|x_f)] - E[\log P(s = Real|x_p)]] - \lambda_2 E[\log P(c = \tilde{y}|x_f)]$$

**Training surrogate models $S$ and $S_{en}$.** As we mentioned in the main paper, to train additional surrogate models $S$ and $S_{en}$, we create a new synthetic dataset generated by our T-ACGAN. Specially, we generate images using 500 pseudo labels for each class. These images are then labeled by the target classifier $T$. To train $S$ and $S_{en}$, we use SGD optimizer with learning rate $lr = 0.1$, momentum 0.9 and weight decay $5 \times 10^{-4}$, and apply the CosineAnnealingLR scheduler [21].

**Inversion.** To reconstruct the images, after training the surrogate model $S$, in the main experimental results, we apply PLGMI [6] as the white-box MI attack on $S$ using our T-ACGAN. For this reconstruction, we use Adam optimizer with the learning rate $lr = 0.002$ and optimize in 600 iterations as [6]. For other experiments, when using KEDMI and GMI as white-box MI attacks on $S$, following [5], we use SGD optimizer with learning rate $lr = 0.02$ and optimize in 2400 iterations.

### E.3 User study

The user interface is shown in Figure E.5. The results are included in the main paper.

## F Related works

Model Inversion (MI) aims to extract/ reconstruct the private information about the training data through a trained model. Depending on the level of information that can be accessed, MI attacks can be classified into three distinct categories: white-box attacks, black-box attacks, and label-only attacks.

**White-Box MI Attack.** In white-box attacks, the attacker is assumed to have complete access to the target model including model weights. Therefore, the MI attack is usually formulated as optimizing an identification loss:

$$x^* = \arg\min_x \mathcal{L}_{id}(x; y, T) \tag{1}$$

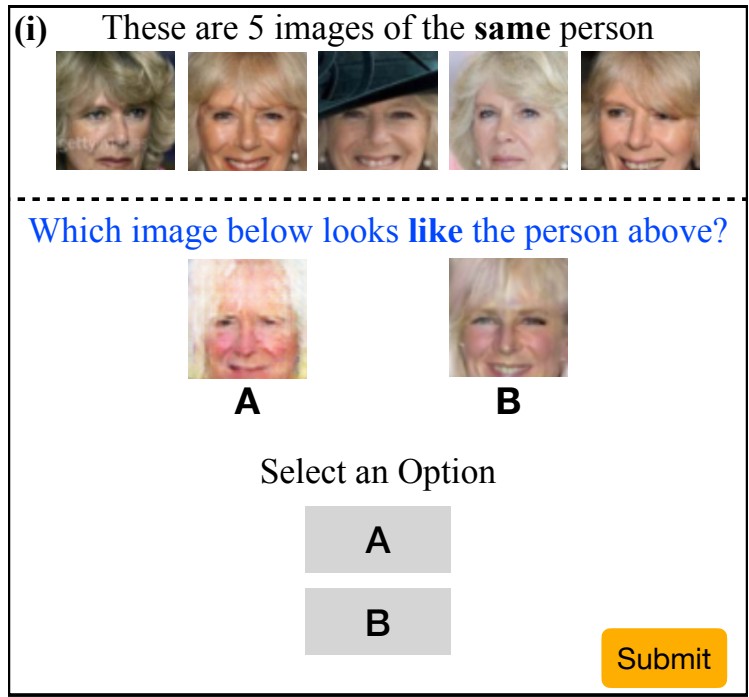

Figure E.5: **Human study setup/ user interface**: We follow the setup proposed by An *et al.* for human study. • In this setup, users are shown 5 real images of a person (identity) as reference. • Then users are required to compare the 5 real images with two inverted images: one from our method, the other from BREPMI. In the above example, A and B correspond to BREPMI and Ours respectively. The order is randomized for each task. Each user is given a maximum of 60 seconds per task, and each task is assigned to 10 unique users. Following [22], we randomly select 50 identities, resulting in 1000 pairs. We use Amazon Mechanical Turk service (MTurk). We use $D_{priv}$ = CelebA, $D_{pub}$ = CelebA, $T$ = FaceNet64.

where $\mathcal{L}_{id}(x; y, T) = -\log \mathbb{P}_T(y|x)$, with $\mathbb{P}_T(y|x)$ denoting the probability (soft label) that the target model $T$ classifies input $x$ as label $y$. When handling a high-dimensional input data $x$ like an image, performing the optimization (Eqn. 1) in input space ends up with degraded results [23, 7]. To overcome this issue, recent white-box approaches [7, 5, 24] constraint the search space into the manifold of related public images using a GAN. More specifically, GMI [7] proposes to train a GAN on a public dataset $\mathcal{D}_{pub}$, and perform the inversion step on the latent space of GAN:

$$z^* = \arg \min_z \mathcal{L}_{id}(z; y, T) + \lambda \mathcal{L}_{prior}(z) \tag{2}$$

where $z^*$ denotes the optimal latent code which is later used by GAN to generate the reconstructed sample, i.e., $x^* = G(z^*)$. In addition, $\mathcal{L}_{prior} = -D(G(z))$ measures the realness of the generated sample. KEDMI [5] improves GMI by introducing inversion-specific GAN, and restoring a distribution of latent space instead of an optimal point. In addition, VMI [24] defines the variational inference in latent space. PLGMI [6] uses the target classifier to produce pseudo label for public data and trains a conditional GAN (cGAN) to limit the search space.

**Black-Box MI Attack.** In the black-box setup, the attackers have access to only model's output and confidence scores (soft labels) which is very limited compared to the white-box setup. Due to this limitation, performing optimization discussed in Eqn. 1, and 2 become unfeasible in the black-box setup. Yang et al. [25] train an inversion model of the target model which serves as an encoder model specifically trained to produce the predicted score (soft labels). Simultaneously, the generator (decoder) is trained to generate the target image based on the predicted score of the inversion model.

**Label-Only MI Attack.** Label-only MI attack relies solely on the final decision of the model, i.e., the predicted label, without any additional information about the model or the confidence score of the prediction. Kahla et. al [1] propose Boundary-Repelling Model Inversion (BREP-MI) to address the model inversion attack under label-only setup. Beginning by initializing a random point that is

already classified into the target class, BREPMI evaluates the model's predicted labels based on other neighbor points in the latent space and estimate the direction to reach the target class's centroid.

In future work, we hope to explore different aspects of model inversion including multimodal learning, advanced knowledge transfer, data-centric applications and different types of generative models [26, 27, 28, 29, 30, 31, 32].

# G   Additional information for checklist

**Amount of Compute.** The amount of compute in this project is reported in Table G.1. We follow NeurIPS guidelines to include the amount of compute for different experiments along with $CO_2$ emission.

Table G.1: Amount of compute in this project. The GPU hours include computations for initial explorations / experiments to produce the reported values. CO2 emission values are computed using `https://mlco2.github.io/impact/`

| Experiment | Hardware | GPU hours | Carbon emitted in kg |
|---|---|---|---|
| Main paper : Table 3 (Repeated 3 times) | RTX A5000 | 306 | 29.56 |
| Main paper : Table 2 and Table 4 | RTX A5000 | 50 | 4.83 |
| Main paper : Figure 1 / Figure 2 | RTX A5000 | 4 | 0.39 |
| Supplementary : All additional analysis/ Ablation study | RTX A5000 | 10 | 0.97 |
| Additional Compute for Hyper-parameter tuning | RTX A5000 | 24 | 2.32 |
| **Total** | | **394** | **38.07** |

**Standard deviation of our experiments (Error Bars).** We report the standard deviation of MI Attack accuracies for 2 experiment setups: • We use $\mathcal{D}_{priv}$ = CelebA [3], $\mathcal{D}_{pub}$ = CelebA [3], $T$ = FaceNet64. • We use $\mathcal{D}_{priv}$ = CelebA [3], $\mathcal{D}_{pub}$ = FFHQ [4], $T$ = FaceNet64. We repeated the entire training and experiments three times. For each trial, we trained T-ACGAN and surrogate models from scratch using different random seeds. The results are shown in Table G.2.

Table G.2: We report standard deviations for MI Attack accuracies for 2 experiment setups over 3 independent runs. The setups include: • We use $\mathcal{D}_{priv}$ = CelebA [3], $\mathcal{D}_{pub}$ = CelebA [3], $T$ = FaceNet64. • We use $\mathcal{D}_{priv}$ = CelebA [3], $\mathcal{D}_{pub}$ = FFHQ [4], $T$ = FaceNet64. We also report the standard deviations for existing SOTA [1].

| Setup | Attack | | Attack acc. ↑ | KNN dt. ↓ |
|---|---|---|---|---|
| $T$ = FaceNet64 | BREPMI | | $74.87 \pm 4.17$ | $1286.04 \pm 1.42$ |
| $\mathcal{D}_{priv}$ = CelebA | | $C \circ D$ | $80.80 \pm 4.35$ | $1305.97 \pm 6.50$ |
| $\mathcal{D}_{pub}$ = CelebA | LOKT | $S$ | $91.96 \pm 2.62$ | $1211.15 \pm 17.06$ |
| | | $S_{en}$ | $93.11 \pm 2.69$ | $1193.16 \pm 25.99$ |
| $T$ = FaceNet64 | BREPMI | | $41.91 \pm 5.09$ | $1484.20 \pm 13.21$ |
| $\mathcal{D}_{priv}$ = CelebA | | $C \circ D$ | $44.33 \pm 4.25$ | $1510.34 \pm 5.07$ |
| $\mathcal{D}_{pub}$ = FFHQ | LOKT | $S$ | $58.42 \pm 3.61$ | $1439.02 \pm 13.79$ |
| | | $S_{en}$ | $62.11 \pm 3.66$ | $1426.89 \pm 12.73$ |