# OpenReview forum: "Label-Only Model Inversion Attacks via Knowledge Transfer"
_NeurIPS.cc/2023/Conference — NeurIPS 2023 poster_

### Official Review · Reviewer_5AoF · 2023-06-20

**Soundness:** 3 good
**Presentation:** 2 fair
**Contribution:** 3 good
**Rating:** 6
**Confidence:** 5

**Summary:**

The paper proposes a novel label-only model inversion attack (MIA). The underlying idea is to train a surrogate model and then perform existing white-box MIAs to reconstruct characteristic samples from the target model's training data. Instead of using the target model to label a public dataset, the paper proposes to train a modified ACGAN using the target model to provide hard labels on the generated samples during training. To demonstrate that the surrogate model assigns similar scores to inputs as the target model does, an analysis of the behavior of both models for easy- and hard-to-classify samples is performed. The proposed approach is then compared to BREP-MI as label-only MIA on different facial image datasets.

**Strengths:**

- The paper proposes a novel label-only MIA, which is the hardest setting for model inversion. The idea of training a surrogate model to perform white-box MIAs is, at least in the model inversion setting, novel.
- The results compared to BREP-MI are promising and demonstrate the attack improvement in the investigated setting.
- The analysis based on [Arpit et al.] for the training behavior and the comparison between the target and surrogate model is quite interesting.

**Weaknesses:**

- While the evaluation procedure follows the one of BREP-MI, I feel that the setting is a bit outdated, given that rather old/small models are investigated as target models. Also, the image size of 64x64 is quite small. It would, therefore, be interesting to see how the attacks perform on images of higher resolution, e.g., 256x256, and other, more recent architectures, e.g., VITs.
- While attack accuracy and feature distance are commonly used in MIA literature, I am not sure if the evaluation should only focus on these metrics. I could imagine that the attacks also craft transferable adversarial examples, which might only leak limited information about the training data but rather exploit adversarial patterns to "fool" the target model - and possibly also the evaluation model. I think another metric would there be helpful, e.g., a CLIP zero-shot prediction, face distance computed by a face verification system, etc.
-It is unclear, how the qualitative samples are selected. Are those cherry-picked? For a fair evaluation, the paper should also state unfiltered attack results (e.g., in the appendix).
- Generally, I think the overall writing should be improved. Whereas the different concepts/ideas are most of the time clearly described, the written text is not always fluently readable.

Small remarks:
- Shortly introduce ACGAN in the introduction. Not every ready may be familiar with the architecture/concept.
- Fig. 1 c) I think there is a small error, the bottom of the third column should also show a green check symbol, right?

**Questions:**

- BREP-MI uses a different GAN architecture (simple DCGAN) compared to ACGAN. What is the general image quality of the two GANs? Could it be the case that the proposed approach also performs better because of higher image quality?
- Why use KEDMI for the white-box attack instead of, e.g., LOMMA [Nguyen et al., CVPR 2023], VMI [Wang et al., NeurIPS 2021], or Plug&Play Attacks [Struppek et al., ICML 2022]? Also, [Struppek et al.] found the attack to be rather unstable in other settings.
- Why choose DenseNet as the surrogate architecture? Does the approach perform comparably to other choices of architecture?

**Limitations:**

Limitations are briefly discussed at the end of the paper. However, I think should be a larger discussion on the influence of the availability of public data, the influence of the GAN architecture, possible limitations in other domains or with higher resolution, etc.

While the discussion on potential negative impact is rather shallow, I think it is sufficient for the paper, also compared to previous work.

---

> ### Author Rebuttal · Authors · 2023-08-10
>
> >It would, therefore, be interesting to see how the attacks perform on images of higher resolution and other, more recent architectures,
>
> We appreciate the Reviewer’s suggestion. We report additional experiment results in 2 parts: (A) The MI attack results using recent architecture ConvNeXt [Liu et. al, CVPR2020] and (B) The MI attack results using ConvNeXt on high-resolution images (224x224).
>
> **A. Attacking recent architectures.**
> We conducted an experiment using ConvNeXt target model (CelebA setup, 1000 classes, 84.85% natural accuracy). Following the exact procedure in the main paper, we perform attacks using BREPMI and our proposed method. Here we use $D_{pub}$ = CelebA. The results are as follow:
> | Attacks | Top 1 ($\uparrow$) | KNN dist ($\downarrow$) |
> |-|-|-|
> | BREPMI | 54.27 $\pm$ 5.36 | 1394.08 |
> | Ours ($C \circ D$) | 64.13 $\pm$ 5.01 | 1358.58 |
> | Ours ($S$) | 81.53 $\pm$ 3.43 | 1270.50 |
> | Ours ($S_{en}$) | 84.00 $\pm$ 2.37 | 1246.30 |
>
> The results show that our proposed method achieves better results in both Top-1 attack accuracy and KNN distance compared to the SOTA BREPMI.
>
> **B. Attacking recent architecture with high-resolution images.**
> Following PPA setup [Struppek et. al. ICML 2022] for high-resolution images, we train a new target classifier, ConvNeXt, using FaceScrub dataset at 224x224 resolution (natural accuracy = 97.26%).
> Given PPA's success in high-resolution whitebox attacks, we use PPA attack in this study. For BREPMI, we use the same StyleGAN used by PPA. The results are as follows:
> | Attacks | Top 1 ($\uparrow$) | Top 5 ($\uparrow$) |
> |-|-|-|
> | BREPMI | 0.00 | 0.94 |
> | Ours ($S_{en}$) | 4.00 | 12.11 |
>
> Both BREPMI and our approach face significant challenges in reconstructing private training data under high-resolution setup.
>
> [PPA] Struppek, L. et al. Plug & Play Attacks: Towards Robust and Flexible Model Inversion Attacks. ICML 2022.
> [ConvNext] Liu, Z. et al. A convnet for the 2020s. CVPR 2020.
>
> $ $
>
>
> > While attack accuracy and feature distance are commonly used in MIA literature, I am not sure if the evaluation should only focus on these metrics.
>
> We appreciate the reviewer's insightful comment regarding the large focus on objective metrics to evaluate MI attacks.
>
> To address this, we go beyond objective metrics and consider **subjective evaluation** of MI attacks. Specifically, we conducted a **human study** to understand the efficacy of our proposed method compared to BREPMI. We follow the setup by An et al. [MIRROR] for human study. Specifically, in this study, • Users are shown 5 real images of a person (identity) as reference. • Then users are required to compare the 5 real images with two inverted images: one from our method, the other from BREPMI. We use $D_{priv}$ = CelebA, $D_{pub}$ = CelebA and $T$ = FaceNet64. Following An et al., we randomly selected 50 identities with 10 unique users evaluating each task accounting to 1000 comparison pairs. More details in the rebuttal PDF submission.
>
> **User study results.** Our human study reveals that users distinctly favor our approach, with **64.30% user preference for images reconstructed using our proposed approach**, in contrast to BREPMI’s lower 35.70% user preference. This subjective evaluation further shows the efficacy of our proposed method in the challenging label-only MI setup.
>
> $ $
>
>
> > It is unclear, how the qualitative samples are selected.
>
> In Supp (line 9), we have already provided the link to **all inverted images** and private images. For each identity/class, we provide the best pair (consisting of an inverted image and a private image) with the smallest KNN distance. We also emphasize that for human study, we **randomly** selected 50 identities and the corresponding samples.
>
> $ $
>
>
> > Generally, I think the overall writing should be improved.
> > Shortly introduce ACGAN in the introduction.
>
> Thank you for the feedback, we will improve the writing accordingly.
>
> $ $
>
>
> > Fig. 1 c) I think there is a small error, the bottom of the third column should also show a green check symbol, right?
>
> Yes, it is indeed a typo. Thank you for pointing out this. We will fix it in the next version.
>
> $ $
>
>
> > BREP-MI uses a different GAN architecture (simple DCGAN) compared to ACGAN.?
>
> For a fair comparison with BREPMI, in this part, we provide the experiment results by training a new T-ACGAN using the **same** architectures as the GAN used by BREPMI. For the discriminator (D), we apply max pooling and add a linear layer before the last layer for the classifier head. As for the generator (G), we retain the same architecture and replace batch normalization with conditional batch normalization.
> The results are shown below:
> | Attacks | Top 1 ($\uparrow$) | KNN dist ($\downarrow$) |
> |-|-|-|
> | BREPMI | 73.93 $\pm$ 4.98 | 1284.41 |
> | Ours ($C \circ D$) | 86.40 $\pm$ 3.18 | 1329.86 |
> | Ours ($S$) | 95.53 $\pm$ 2.42 | 1199.93 |
> | Ours ($S_{en}$) | 97.53 $\pm$ 1.31 | 1168.20 |
>
> Here, we use $T$ = FaceNet64, $D_{priv}$ = CelebA, and $D_{pub}$ = CelebA. Our results are better than BREPMI when using the same GAN architecture.
>
> $ $
>
> > Why use KEDMI for the white-box attack.
>
> We have used a range of white-box attacks, including PLGMI [Yuan et. al. AAAI 2023] and KEDMI. Our main results use PLGMI due to its superior white-box MI attack performance compared to other state-of-the-art methods.
> As for PPA [Struppek et. al. ICML 2022], both our method and BREPMI struggle to effectively reconstruct high-resolution images potentially due to large StyleGAN latent space. We leave it for our future work.
>
> $ $
>
> > Why choose DenseNet as the surrogate architecture? Does the approach perform comparably to other choices of architecture?
>
> In section B.4 (supplementary), we study the effect of model architectures on model accuracy and MI attack accuracy to empirically justify our use of DenseNet model variants [10] as surrogate models.
>
> $ $
>
> > Limitations:
>
> Thank you for the comment. We will include extended discussion on limitations.

---

> > ### Comment · Reviewer_5AoF · 2023-08-11
> >
> > I thank the authors for their clarification and additional insights. I also appreciate their effort to answer my questions and run additional experiments with more metrics.
> >
> > Furthermore, I like that the authors conducted a human study on the preferences of the attack results. However, if I understand it correctly, the study only allowed its participants to choose between the results of BREP-MI and the proposed attack, and provides no option to reject both results if the attacks fail. This would have made the study even more valuable.
> >
> > While I see the additional experiments on the 64x64 scale demonstrate that the approach indeed beats the existing BREP-MI attack, I also recognize the limited effectiveness of the proposed approach on larger images. This limits the attack significantly in my view.
> >
> > Overall, I will keep my initial score.

---

> > > ### Author Response · Authors · 2023-08-13
> > > **Additional human study and experiments on 128x128 label-only MI attacks**
> > >
> > > > Furthermore, I like that the authors conducted a human study on the preferences of the attack results. However, if I understand it correctly, the study only allowed its participants to choose between the results of BREP-MI and the proposed attack, and provides no option to reject both results if the attacks fail. This would have made the study even more valuable.
> > >
> > > We thank the reviewer for the insightful feedback. For human study, we remark that we strictly followed the framework by An et al. [MIRROR] for a fair setup. The goal was to verify that our method is preferable compared to BREPMI under human evaluation.
> > >
> > > Nevertheless, we respect the reviewer’s request and conducted an **additional human study** following the exact setup of An et. al [MIRROR], but including the **option of rejecting both results** in cases where both reconstructed images fail to capture the target identity. The additional human study consistently supports that our method is superior compared to BREPMI. Details of the study can be found below. We will include the study and finding in the revised submission.
> > >
> > > **Additional User Study setup.**
> > >
> > > We follow the exact experiment setup as our rebuttal experiment.  $D_{priv}$ = CelebA, $D_{pub}$ = CelebA and $T$ = FaceNet64. Following An et al., we randomly selected 50 identities with 10 unique users evaluating each task accounting to 1000 comparison pairs. We used Amazon Mechanical Turk (MTurk) for these experiments.
> > >
> > > **Additional User Study results.**
> > >
> > > Our human study reveals that **only 2.44% of attack evaluations are rejected by the users (Choose “Reject Both” option)**, with remaining users distinctly favoring our approach. In particular,  **60.98% users prefer images reconstructed using our proposed approach**, in contrast to BREPMI’s lower 36.59% user preference. The results are included below.
> > >
> > > | Method | User Preference ($\uparrow$) |
> > > |-|-|
> > > | BREPMI | 36.59% |
> > > | Ours ($S_{en}$) | **60.98%** |
> > > | Reject Both | 2.44% |
> > >
> > > We hope our additional human study adequately addresses the reviewer’s concerns. We are happy to address any additional concerns during this discussion period.
> > >
> > > [MIRROR] An, Shengwei et al. MIRROR: Model Inversion for Deep Learning Network with High Fidelity. Proceedings of the 29th Network and Distributed System Security Symposium.
> > >
> > > $ $
> > > > While I see the additional experiments on the 64x64 scale demonstrate that the approach indeed beats the existing BREP-MI attack, I also recognize the limited effectiveness of the proposed approach on larger images. This limits the attack significantly in my view.
> > >
> > > We agree with the reviewer's observation on larger images. Our previous results suggest that 224x224 is too challenging for label-only model inversion. Therefore, we have conducted another experiment on **128x128 resolution**. Details can be found below. The results suggest that label-only attack is feasible at 128x128. Furthermore, the **results for larger images (128 x 128) consistently support that our method is superior compared to BREPMI**.
> > >
> > > It's worth mentioning that the concept of label-only model inversion attack is very new, and the recent work, BREPMI, stands as the singular effort in this challenging area. Furthermore, compared to white-box/black-box MI, label-only MI is the most restrictive setting and is very challenging. We believe our study can provide new insight on the effectiveness of SOTA label-only attack at a higher resolution of 128x128, paving the way to future label-only model inversion attacks at resolutions beyond 128x128.
> > >
> > > **Experiment setup.**
> > >
> > > We conduct the experiment using $T$ = Resnet-152 trained on CelebA dataset with the resolution **128x128**. The natural accuracy is 86.07%.
> > >
> > > For fair comparison between BREPMI and our proposed method, T-ACGAN has the **same** GAN architectures used by BREPMI. For the discriminator (D), we apply max pooling and add a linear layer before the last layer for the classifier head. As for the generator (G), we retain the same architecture and replace batch normalization with conditional batch normalization. We use $D_{pub}$ = CelebA.
> > >
> > > **Results.**
> > >
> > > Our proposed method outperforms BREPMI, with top 1 accuracy surpassing BREPMI by **20.27%**. Our inverted images are closer to private training samples than BREPMI (smaller KNN distance).
> > >
> > > Table. Attacking with high-resolution images. T = ResNet152, $D_{priv}$ = CelebA,  $D_{pub}$ = CelebA, image size = 128x128.
> > >
> > > | Attacks | Top 1 ($\uparrow$) | KNN dist ($\downarrow$) |
> > > |-|-|-|
> > > | BREPMI | 50.33 ± 4.71 | 1389.09 |
> > > | Ours ($S_{en}$) | **70.60 ± 4.43** | **1320.16** |
> > >
> > > We hope Reviewer could consider to increase the ratings from initial decision if the additional set of experiments can resolve Reviewer’s concern.

---

> > > > ### Comment · Reviewer_5AoF · 2023-08-14
> > > >
> > > > Thank you for updating the user study and running experiments on larger resolutions. While I am still not fully convinced due to the limitation of image resolution, I acknowledge the authors' effort in the rebuttal. I will, therefore, raise my score.

---

> > > > > ### Author Response · Authors · 2023-08-15
> > > > > **Thank you for increasing your recommendation**
> > > > >
> > > > > We sincerely thank the reviewer for the insightful comments. Thank you for the positive feedback and for increasing the rating. We will include all additional results in the revised version.
> > > > > Sincerely,
> > > > > Authors

---

### Official Review · Reviewer_txcd · 2023-06-24

**Soundness:** 2 fair
**Presentation:** 3 good
**Contribution:** 2 fair
**Rating:** 5
**Confidence:** 4

**Summary:**

This paper proposes a label-only model inversion (MI) attack. It focuses on a more challenging scenario where the target network only outputs the label. It trains a class-conditional GAN based on some public data and the label information provided by the target network. A surrogate network can be trained at the same time as training the GAN or using the fake data generated by the trained GAN. Then they use the existing white-box MI attack method on the surrogate network to generate the images. The generated images are used to complete the label-only attack on the target model. Experiments conducted on three datasets and models show its effectiveness in terms of attack accuracy and feature similarity with the ground truth image.

**Strengths:**

1. This paper targets a more challenging MI attack scenario where the subject model only returns the labels.
2. Experimental results show the proposed method outperforms one existing baseline.
3. Code is provided.

**Weaknesses:**

1. Some critical details of the evaluation are missing. What are the target labels? How are they selected? When splitting the dataset into private and public datasets, is the separation based on identities/labels? That is, are there overlapped labels in the private and public datasets?

2. It's not clear how this method can handle the rare labels (similar to the class imbalance issue mentioned in this paper). If T has 1K classes in the private dataset, I wonder if the synthesized images in eq 4 can equally cover all the 1K. If not, how can the method attack those uncovered labels?

3. The evaluated models and datasets are not large enough. It would be more persuasive if the method can attack models trained on larger datasets with more classes. For example, ResNet50 trained on VGG Face2, and Inception Resnet on CASIA.

4. User studies are necessary to show the leakage of privacy brought by the inversion results. For example, show the inversion result of one target label and images of the target label and other several labels to users and check if they can recognize the target label. Similarly, show one image of the target label and the inversion results of the target label and other labels.

5. Line 188. Based on the current definitions of P1 and P2, they are equivalent. Use $Pr(x)$ denotes the probability of sample $x$ has high $P_T$. P1 means if $x$ has high $P_S$, then $Pr(x)$ is high. P2 means if $x$ has high $P_S$, then $1-Pr(x)$ is low. They are describing the same thing. However, according to line 212, P2 should be it's uncommon for low-likelihood samples under $T$ to have high likelihood under $S$.

6. Another defense method [MID] should be evaluated.

7. Missing related work similar to cite [PLUG] [MIRROR].

[MID] Wang, Tianhao, et al. “Improving Robustness to Model Inversion Attacks via Mutual Information Regularization.” AAAI 2021.

[PLUG] Struppek, Lukas, et al. "Plug & Play Attacks: Towards Robust and Flexible Model Inversion Attacks." ICML 2022.

[MIRROR] An, Shengwei, et al. "Mirror: Model inversion for deep learning network with high fidelity." NDSS 2022.

**Questions:**

1. What are the target labels in the evaluation? How are they selected?

2. How can the method attack target labels that are not sampled during training TACGAN?

3. Are there overlapped identities/labels in the private and public datasets?

4. Can the proposed method successfully attack larger models and datasets?

5. Can humans correctly recognize the inversion results and the target labels?

6. Is the proposed method robust against [MID]?

**Limitations:**

Possible limitations are 1) high time cost, 2) about 13 million required queries, and 3) uncommon labels.

---

> ### Author Rebuttal · Authors · 2023-08-10
>
> > What are the target labels? Are there overlapped identities/labels in the private and public datasets?
>
> Pls see Ln 223: “To ensure a fair comparison, when evaluating our method, we use the **exact** same experimental setup as BREPMI [6].” Following [6], each dataset is divided into two distinct/ non-overlapping sets: private/ public dataset. There are no shared identities/ labels.
>
> $~$
> > If T has 1K classes in the private dataset, I wonder if the synthesized images in eq 4 can equally cover all the 1K.
>
> Yes, our method can achieve adequate coverage for all 1K classes (see experiment below). Our decision knowledge transfer using target model enables generation of diverse conditional samples using our T-ACGAN (Eqn. 3, 4) adequately covering all 1K classes.
>
> **Experiment setup.** Let $x = G(z, y)$, $\hat{y} = T(x)$. For each class $y$, $\gamma$ represents the percentage of samples with y the same as $\hat{y}$ (Supp. A.2). We use $T$ = FaceNet64, $D_{priv}$ = CelebA, $D_{pub}$ = CelebA setup to demonstrate the $\gamma$ distribution for all 1K classes using our T-ACGAN.
>
> **Results.** The results are included in rebuttal PDF submission. As one can observe, synthesized images using T-ACGAN renders adequate coverage for all 1K classes with minimum $\gamma$ value being 6.6%.
>
> $~$
> > The evaluated models/ datasets are not large enough.
>
> We emphasize we **strictly** follow BREPMI [6] for model/dataset setups. Nevertheless, we conduct additional experiments. We use target classifier $T$ = ResNet50, which is trained on the $D_{priv}$ = VGGFace2 dataset consisting of 3.3 million images / 8631 training classes. We use $D_{pub}$ = FFHQ. We use E= Inception-V1 trained on VGGFace2.
> We report the result in Table 1 [txcd]. Our method significantly improves attack accuracy and KNN distance compared to BREPMI, with top1 accuracy surpassing BREPMI by 12.6%-19.8%.
> Table 1 [txcd]. Attack results. We use $T$ = ResNet50, $D_{priv}$ = VGGFace2, $D_{pub}$ = FFHQ, $E$ = Inception-V1.
> | Attacks | Top 1 ($\uparrow$) | KNN dist ($\downarrow$) |
> |-|-|-|
> | BREPMI | 56.00 $\pm$ 4.65 | 428.19 |
> | Ours ($C \circ D$) | 68.60 $\pm$ 3.98 | 389.28 |
> | Ours ($S$) | 71.20 $\pm$ 3.03 | 397.37 |
> | Ours ($S_{en}$) | 75.80 $\pm$ 2.01 | 379.50 |
>
> $~$
> > User studies are necessary to show the leakage of privacy brought by the inversion results.
>
> > Can humans correctly recognize the inversion results and the target labels?
>
> We conducted a **human study** to reveal serious privacy leakage under our proposed method compared to BREPMI. We follow the setup proposed by An et al. [MIRROR] for human study. Specifically, in this study, • Users are shown 5 real images of a person (identity) as reference. • Then users are required to compare the 5 real images with two inverted images: one from our method, the other from BREPMI. We use $D_{priv}$ = CelebA, $D_{pub}$ = CelebA and $T$ = FaceNet64. Following An et al., we randomly selected 50 identities with 10 unique users evaluating each task accounting to 1000 comparison pairs. More details on human study, user interface, example, experiment details and results are in the PDF submission.
>
> **User study results.** Our human study reveals that users distinctly favor our approach, with **64.30% user preference** for images reconstructed using our proposed approach, in contrast to BREPMI’s lower 35.70% user preference. Detailed results included in PDF submission. This further shows the efficacy of our method.
>
> $~$
> > Line 188. Based on the current definitions of P1 and P2, they are equivalent.
>
> Thank you, we will refine this discussion. In fact, P1 could justify our method.
>
> $~$
> > Another defense method [MID] should be evaluated.
> > Is the proposed method robust against [MID]?
>
> Following reviewer’s suggestion, we use the official implementation to train MID on CelebA setup (Natural Acc.=79.16%). We use $T$ = MID, $D_{priv}$ = CelebA, and $D_{pub}$ = CelebA. The results are as follows:
> | Attacks | Top 1 ($\uparrow$) | KNN dist ($\downarrow$) |
> |-|-|-|
> | BREPMI | 39.20 $\pm$ 4.19 | 1458.61 |
> | Ours ($C \circ D$) | 44.13 $\pm$ 3.54 | 1475.73 |
> | Ours ($S$) | 55.33 $\pm$ 4.40 | 1393.76 |
> | Ours ($S_{en}$) | 60.33 $\pm$ 4.76 | 1374.34 |
>
> Our proposed method is robust/ surpasses BREPMI's performance across all setups. These results are comparable to the SOTA defense BiDO-HSIC (Table 4-main paper, Table B.2-Supp).
>
> [MID] Wang, Tianhao, et al. “Improving Robustness to Model Inversion Attacks via Mutual Information Regularization.” AAAI 2021.
>
> $~$
> > Missing related work similar to cite [PLUG] [MIRROR].
>
> We will update related works.
>
> $~$
> > What are the target labels in the evaluation? How are they selected?
>
> See Ln 223: “To ensure a fair comparison, when evaluating our method, we use the exact same experimental setup as BREPMI [6].” Following [6], we attack the first 300 out of 1000 labels in the experiments using CelebA dataset.  In cases of Facescrub and Pubfig, we attack all the labels of the target classifier (200 and 50, respectively).
>
> $~$
> > How can the method attack target labels that are not sampled during training T-ACGAN?
>
> This highlights a critical concern with the SOTA label-only methods. Unlike BREPMI, our T-ACGAN successfully addresses the challenge of covering all the labels enabled by decision knowledge transfer from the target model. Our results demonstrate the effectiveness of T-ACGAN, as it covers all 1,000 classes adequately.
>
> $~$
> > Can the proposed method successfully attack larger models/ datasets?
>
> Yes, under the setup with $T$=ResNet50, $D_{priv}$=VGGFace2 and $D_{pub}$=FFHQ, our proposed method outperforms BREPMI, with top 1 accuracy surpassing BREPMI by **12.6%-19.8%**.
>
> $~$
> > Possible limitations: 1) high time cost, 2) ~13 million required queries, 3) uncommon labels.
>
> Table 5 (main paper) shows that our method can reduce required queries by more than 30% compared to SOTA BREPMI. The reviewer's insight is greatly appreciated, and we agree that further reduction is desirable.

---

> > ### Comment · Reviewer_txcd · 2023-08-21
> >
> > Thank the authors for the response. Thanks for the extra evaluation. It resolves some of my concerns and raises my score a bit.
> >
> > It's interesting to see the randomly sampled images from a GAN can cover "all 1K classes with the minimum value being 6.6%." I wonder for the test labels, what are the $\gamma$ values? Also, for the new VGGFace2 case, what does the $\gamma$ distribution look like, what labels are selected, and what are their $\gamma$ values?
> >
> > Besides, to address my question
> >
> > > What are the target labels? Are there overlapped identities/labels in the private and public datasets?
> >
> > Instead of saying
> > > Pls see Ln 223: “To ensure a fair comparison, when evaluating our method, we use the exact same experimental setup as BREPMI [6].” Following [6], each dataset is divided into two distinct/ non-overlapping sets: private/ public dataset. There are no shared identities/ labels.
> >
> > I suggest including the details in the paper to make it more self-contained, instead of asking readers to look for another paper to see how this paper's experiment is set up.

---

> > > ### Author Response · Authors · 2023-08-21
> > > **Thank you for your positive feedback**
> > >
> > > > Thank the authors for the response. Thanks for the extra evaluation. It resolves some of my concerns and raises my score a bit.
> > >
> > > We thank the Reviewer for your time and positive feedback.
> > >
> > > Remark: Since external links are not allowed in the response, **we have included the additional requested figures in the PDF file named ‘Reviewer_txcd_Aug21_response.pdf' in the Google Drive link**. This anonymized Google Drive link was already shared with the Area Chair earlier.
> > >
> > > $ $
> > > > It's interesting to see the randomly sampled images from a GAN can cover "all 1K classes with the minimum value being 6.6%." I wonder for the test labels, what are the $\gamma$ values?
> > >
> > > We strictly follow previous work [PLGMI] to select the first 300 classes of the target classifier to test the attack methods. Following reviewer suggestion, we analyze these 300 classes and observe the minimum value of $\gamma$ of 28.1%. The histogram of $\gamma$ for these 300 classes can be found in Fig. D1 (Reviewer_txcd_Aug21_response.pdf).
> > >
> > > $ $
> > > > Also, for the new VGGFace2 case, what does the gamma distribution look like, what labels are selected, and what are their $\gamma$ values?
> > >
> > > We strictly follow previous work [MIRROR] in selecting target attack labels. In particular, we select the first 100 classes of 8631 classes to attack and evaluate the results.
> > >
> > > Regarding the $\gamma$ distribution, there is a significant distribution shift between $D_{priv} = \text{VGGFace2}$ and $D_{pub} = \text{FFHQ}$, and our TACGAN is trained using $D_{pub} = \text{FFHQ}$. Consequently, there is a small number of classes that have $\gamma = 0%$, i.e. only  10% of the selected labels. Interestingly, we observe that **our TACGAN still demonstrates the capability to cover all classes**, and all classes have at least 1 image. This happens because images generated by other pseudo-labels $y \neq k$ could be classified into the class $y = k$ by the target classifier. Therefore, the number of images per class can be higher than $\gamma%$. The histogram of $\gamma$ and the number of image per class can be found in Fig. D2 and Fig. D3 (Reviewer_txcd_Aug21_response.pdf).
> > >
> > > [PLGMI] Yuan, X. et al. Pseudo Label-Guided Model Inversion Attack via Conditional Generative Adversarial Network. AAAI 2023.
> > >
> > > [MIRROR] An, Shengwei et al. MIRROR: Model Inversion for Deep Learning Network with High Fidelity. Proceedings of the 29th Network and Distributed System Security Symposium, 2022.
> > >
> > >
> > > $ $
> > >
> > > > I suggest including the details in the paper to make it more self-contained, instead of asking readers to look for another paper to see how this paper's experiment is set up.
> > >
> > > Thank you for the feedback, we will make the paper more self-contained.

---

### Official Review · Reviewer_Um8m · 2023-07-10

**Soundness:** 3 good
**Presentation:** 2 fair
**Contribution:** 3 good
**Rating:** 5
**Confidence:** 3

**Summary:**

The paper focuses on the setting where the adversary has only label access to the target model. The authors suggested to use transfer learning and generative models to improve the model inversion attacks. In particular the authors proposed T-ACGAN which is a variation of ACGAN that leverages the access to the target model and uses the label that the target model generates.

**Strengths:**


The paper considers a realistic thread model where the adversary does not have whitebox access to the model.
 The authors also evaluate different datasets with different metrics. The attack also reduces the number of required queries.
The authors also evaluated the settings where the public and the private dataset are not the same.



**Weaknesses:**

The details about the target labels and the model that is used to compute the attack accuracy metrics is missing. Structure of the paper can be improved to be more self-contained.

While the work follows the existing evaluation scheme of the previous works, the results are hard to interpret. The attack accuracy metrics mainly relies on the classification accuracy of the generated images which might not be the cause of the privacy leakage. One suggestion can be instead of using the target model feature space, the authors can use some of the open source image embedding models such as CLIP in their evaluation which might make the results easier to understand.


**Questions:**

It would be interesting to also include structural similarity scores for the evaluations.

It would be interesting to compare the knn distances of the attacks with the average distance between the images of the same class. The idea here is to see if the attack only gives an average image from the class or can it give more details. Same for the other metrics.




**Limitations:**

Limitation are addressed in the work.

---

> ### Author Rebuttal · Authors · 2023-08-10
>
>
> > In Fig 1.c The authors show that the adversary does not have access to the hard labels ? is that correct?
>
> Thank you for pointing out this to us. It is a typo, the adversary has access to the hard labels.
>
> $ $
>
> > Looking at the **membership inference attack** literature, the better way of measuring the MIA performance is by looking at false positive true positive curve and not accuracy [R1]. However the work only reports accuracy.
> The authors did not compare their results with existing approaches that use label-only setting[R2]. In particular I think a better baseline would be to compare the results (preferably ROC) using [R2] to estimate the confidence of labels and then use techniques from [R1] to do a hypothesis testing for membership inference attacks. The current comparison is not detailed.
>
> >[R1] Carlini, N., Chien, S., Nasr, M., Song, S., Terzis, A., & Tramer, F. (2022, May). **membership inference attack** from first principles. In 2022 IEEE Symposium on Security and Privacy (SP) (pp. 1897-1914).
>
> >[R2] Choquette-Choo, Christopher A., et al. "Label-only **membership inference attacks**." International conference on machine learning. PMLR, 2021.
>
> Dear Reviewer Um8m,
>
> Our paper focuses on Model Inversion, not Membership Inference as the reviewer suggested. Therefore, we believe your comments are completely irrelevant and out of context.
>
>  **Model Inversion and Membership Inference are totally different** as discussed in many works [a,b,c,d].
> For model inversion, the objective is to **infer and reconstruct the training data of a trained model.**
> For membership inference, the objective is to **determine whether a specific sample was used to train the model or not**.
>
> Nevertheless, **we do appreciate a lot Reviewer's valuable time in reading our submission**. Thank you very much.
>
> [a] Yuheng Zhang, Ruoxi Jia, Hengzhi Pei, Wenxiao Wang, Bo Li, Dawn Song. The Secret Revealer: Generative Model-Inversion Attacks Against Deep Neural Networks. CVPR-2020.
>
> [b] Tanuwidjaja, H. C., Choi, R., Baek, S., & Kim, K. (2020). Privacy-preserving deep learning on machine learning as a service—a comprehensive survey. IEEE Access 2020.
>
> [c] Rigaki, M., & Garcia, S. (2020). A survey of privacy attacks in machine learning. arXiv preprint arXiv:2007.07646.
>
> [d] Yang Ziqi, Shao Bin, Xuan Bohan, Chang Ee-Chien, and Zhang Fan. 2020. Defending model inversion and membership inference attacks via prediction purification. arXiv preprint arXiv:2005.03915 (2020).

---

> > ### Comment · Reviewer_Um8m · 2023-08-14
> >
> > I am really sorry for misunderstanding, and thank you for pointing it out, I changed the review accordingly and change the score to reflect the review.

---

> > > ### Author Response · Authors · 2023-08-15
> > > **[Response for Reviewer Um8m] Part 1**
> > >
> > > > I am really sorry for misunderstanding, and thank you for pointing it out, I changed the review accordingly and change the score to reflect the review.
> > >
> > > We are glad that misunderstanding has been resolved. Reviewer is very responsible and professional to provide us with a new set of reviews, which we appreciate greatly.
> > >
> > > Below, we address all Reviewer’s comments one by one.
> > >
> > > > The details about the target labels and the model that is used to compute the attack accuracy metrics is missing. Structure of the paper can be improved to be more self-contained.
> > >
> > > We thank the reviewer for the suggestion. We will update the description of the target labels and the evaluation model.
> > >
> > > Pls see Ln 223: “To ensure a fair comparison, when evaluating our method, we use the **exact** same experimental setup as BREPMI [6].” Following [6], we attack the first 300 out of 1000 labels in the experiments using CelebA dataset.  In cases of Facescrub and Pubfig, we attack all the labels of the target classifier (200 and 50, respectively). As for the evaluation model, we use FaceNet which is trained on the private dataset and has higher resolution than the target classifier (image resolution 112x112). **We remark that all pre-trained evaluation models are released publicly by BREPMI [6], and we adopt these models in our experiments for fair comparison.**
> > >
> > > We will also improve the structure of the paper as suggested by Reviewer.
> > >
> > > $ $
> > >
> > > > While the work follows the existing evaluation scheme of the previous works, the results are hard to interpret. The attack accuracy metrics mainly relies on the classification accuracy of the generated images which might not be the cause of the privacy leakage. One suggestion can be instead of using the **target model feature space**, the authors can use some of the open source image embedding models such as CLIP in their evaluation which might make the results easier to understand.
> > >
> > > We thank the reviewer for the suggestion. First, we would like to emphasize that we strictly follow experiment setups in BREPMI [6]. To evaluate the attack results, we use the **evaluation model feature space** to measure the attack accuracy and the KNN distance.
> > >
> > > **Lower natural accuracy of CLIP-based facial image classifier suggests that CLIP embedding based evaluation is suboptimal**. As per reviewer’s suggestion, we build a prototype-based classifier using CLIP to verify whether CLIP features are good image embeddings for our face-recognition models. Our results using $T$ = CLIP-RN50, $D_{priv}$ = CelebA, $D_{pub}$ = CelebA show that the natural accuracy of the aforementioned classifier is **only 59.59%** on the private test set suggesting that CLIP features may not be suitable for our study, i.e.: Pre-trained CLIP features (which are self-supervised trained on large but general object dataset) are not discriminative enough to classify human facial images of different identities (different individuals). However, we agree with the reviewer that going beyond objective metrics is beneficial to further interpret the results. Therefore, we conduct human studies to address the Reviewer's comment, details below.
> > >
> > > Natural acc of CLIP-based evaluation model and our evaluation model which has been used in our experiments (We remark that our procedure to build the CLIP-based evaluation model is the same as that in [VMI], but [VMI] uses a specialized facial feature extractor instead of the general CLIP feature extractor.)
> > >
> > > | Evaluation model | Training set | Test set |
> > > |-|-|-|
> > > | CLIP | 67.06% | 59.59% |
> > > | Our evaluation model (FaceNet) which has been used in our experiments  | 99.99% | 95.88% |
> > >
> > > [VMI] Wang, K. C., Fu, Y., Li, K., Khisti, A., Zemel, R., & Makhzani, A. (2021). Variational model inversion attacks. Neurips 2021.
> > >
> > > To address Reviewer’s comment, we go beyond objective metrics and consider **subjective evaluation** of MI attacks. Specifically, we conducted **2 human studies** to understand the efficacy of our proposed method compared to BREPMI. We follow the exact framework of An et al. [MIRROR] for human study. **Both our human studies show that users distinctly favor our approach compared to BREPMI**. The experiment setup and results are included below.
> > >
> > > (cont'd)

---

> > > > ### Author Response · Authors · 2023-08-15
> > > > **[Response for Reviewer Um8m] Part 2**
> > > >
> > > >
> > > > **User study setup.**
> > > >
> > > > **Study 1.** We follow the setup by An et al. [MIRROR] for human study. Specifically, in this study, • Users are shown 5 real images of a person (identity) as reference. • Then users are required to compare the 5 real images with two inverted images: one from our method, the other from BREPMI. We use $D_{priv}$ = CelebA, $D_{pub}$ = CelebA and $T$ = FaceNet64. Following An et al., we randomly selected 50 identities with 10 unique users evaluating each task accounting to 1000 comparison pairs. More details in the rebuttal PDF submission. We used Amazon Mechanical Turk (MTurk) for these experiments.
> > > >
> > > > **Study 2.** Here we follow the exact setup as User Study 1 above, but including the **option of rejecting both results** in cases where both reconstructed images fail to capture the target identity (This variant was suggested by Reviewer 5AoF).
> > > >
> > > > **User Study results.**
> > > >
> > > > **Study 1.** These results are included in rebuttal PDF submission. Our first human study reveals that users distinctly favor our approach, with **64.30% user preference for images reconstructed using our proposed approach**, in contrast to BREPMI’s lower 35.70% user preference.
> > > >
> > > > **Study 2.** Our second human study reveals that **only 2.44% of attack evaluations are rejected by the users (Choose “Reject Both” option)**, with remaining users distinctly favoring our approach. In particular,  **60.98% users prefer images reconstructed using our proposed approach**, in contrast to BREPMI’s lower 36.59% user preference. The results are included below.
> > > >
> > > > | Method | User Preference ($\uparrow$) |
> > > > |-|-|
> > > > | BREPMI | 36.59% |
> > > > | Ours ($S_{en}$) | **60.98%** |
> > > > | Reject Both | 2.44% |
> > > >
> > > > These subjective evaluations further show the efficacy of our proposed method in the challenging label-only MI setup.
> > > >
> > > > [MIRROR] An, Shengwei et al. MIRROR: Model Inversion for Deep Learning Network with High Fidelity. Proceedings of the 29th Network and Distributed System Security Symposium.
> > > >
> > > > $ $
> > > >
> > > > > It would be interesting to also include structural similarity scores for the evaluations.
> > > >
> > > > Thank you for your suggestion. As per reviewer's request, we include the average structural similarity (SSIM) scores for reconstructed images (measured with respect to private training data) for BREPMI and Ours. We also include SSIM scores for LOMMA, the SOTA White-box MI attack method for reference.
> > > >
> > > > We use $T$ = FaceNet64, $D_{pub}$ = CelebA, $D_{priv}$ = CelebA setup to report SSIM results. The results are included below. **These additional SSIM values clearly indicate that our proposed method reconstructs samples with higher structural similarity compared to BREPMI**. We will include all these additional results in the revised version.
> > > >
> > > > | Attack | SSIM ($\uparrow$) |
> > > > |-|-|
> > > > | LOMMA (SOTA White-box MI Attack for reference) | 0.443  ± 0.113 |
> > > > | BREPMI (Label only MI) | 0.372  ± 0.115 |
> > > > | Ours ($S_{en}$) (Label only MI) | **0.416  ± 0.105** |
> > > >
> > > > [LOMMA] Nguyen, N. B., Chandrasegaran, K., Abdollahzadeh, M., & Cheung, N. M. (2023). Re-thinking Model Inversion Attacks Against Deep Neural Networks. In CVPR 2023.
> > > >
> > > > $ $
> > > >
> > > > > It would be interesting to compare the knn distances of the attacks with the average distance between the images of the same class. The idea here is to see if the attack only gives an average image from the class or can it give more details. Same for the other metrics.
> > > >
> > > > Following the reviewer’s suggestion, we compute **Average dist** which  is the average distance between the reconstructed images of a specific identity and the private images of the same identity. Specifically, the average distance is computed using the L2 distance in the feature space which is extracted from the evaluation model’s penultimate layer.
> > > >
> > > > Additionally, we compute **Feature dist** [RLBMI]. Feature dist measures the distance between the features of a reconstructed image and the **centroid** features of the private images of the same target class. Particularly, the features of an image are the output of the penultimate layer of the evaluation model.
> > > >
> > > > The result shows that our reconstructed images are closer to a specific private sample than BREPMI (KNN distance is smaller). Our reconstructed images are also closer to the **average image** of the private samples (Our average dist and feat dist are smaller than BREPMI).
> > > >
> > > > Table. The comparison of KNN distance, Average distance, and Features distance. We use $T$ = FaceNet64, $D_{priv}$ = CelebA, $D_{pub}$ = CelebA.
> > > >
> > > > | Attack | KNN dist ($\downarrow$) | Average dist ($\downarrow$) | Feat Dist ($\downarrow$) |
> > > > |-|-|-|-|
> > > > | BREPMI | 1284.41 | 1719.86 | 1181.66 |
> > > > | Ours | **1181.72** | **1600.68** | **1062.48**  |
> > > >
> > > > [RLBMI] Han, G., Choi, J., Lee, H., & Kim, J. (2023). Reinforcement Learning-Based Black-Box Model Inversion Attacks. CVPR 2023.
> > > >
> > > > We hope Reviewer could consider to increase the ratings from initial decision if the additional set of experiments can resolve Reviewer’s concern. Thank you very much again for the new set of comments.

---

### Official Review · Reviewer_VBSE · 2023-07-14

**Soundness:** 3 good
**Presentation:** 3 good
**Contribution:** 3 good
**Rating:** 6
**Confidence:** 5

**Summary:**

This paper proposes a two-step label-only model inversion attack using knowledge transfer. In the first step, a surrogate model is trained via knowledge transfer. In the second step, a white-box model inversion attack is performed on the surrogate model. The authors explore several approaches to train the surrogate including learning-based model extraction using a public dataset labeled by the target model and the use of a classifier head of the discriminator of a GAN trained on a public dataset labeled by the target model.

**Strengths:**

Several experiments have been performed to evaluate the performance of the proposed solution against different target models.

**Weaknesses:**

The main weakness is the lack of comparison with existing works that also rely on GAN (cf. questions).

**Questions:**

It will be great if the authors can elaborate more on the difference between T-ACGAN and GAMIN [10]. The authors briefly describe GAMIN as an "an adversarial approach for black-box MI". However, the current proposal shares several similarities with GAMIN, namely the use of GANs to learn a surrogate, which is then used to perform the MI attack. Furthermore, the authors classify GAMIN as a black-box MI attack relying on soft labels, while Algorithm 1 in [10] seems to indicate it could be considered a label-only MI.

- How does the intra-class diversity of the dataset affect the result of the attack?

- What explains the difference in performance between ACGAN scenarios described in Section 6.2 and T-ACGAN given they seem to rely on $\widetilde{D}_{pub}$ which is obtained using the target model

[10] Ulrich Aivodji, Sebastien Gambs, and Timon Ther. Gamin: An adversarial approach to black-box model 364 inversion. In The AAAI Workshop on Privacy-Preserving Artificial Intelligence, 2020.

---

> ### Author Rebuttal · Authors · 2023-08-10
>
> > It will be great if the authors can elaborate more on the difference between T-ACGAN and GAMIN [10]. The authors briefly describe GAMIN as an "an adversarial approach for black-box MI". However, the current proposal shares several similarities with GAMIN, namely the use of GANs to learn a surrogate, which is then used to perform the MI attack. Furthermore, the authors classify GAMIN as a black-box MI attack relying on soft labels, while Algorithm 1 in [10] seems to indicate it could be considered a label-only MI.
>
> Thank you for the comment. There are major fundamental differences between **GAMIN (for black-box attack and requires soft prediction)** and **T-ACGAN (our proposed algo for label-only attack and does not require soft prediction)**. See below Table for comparison.
>
> We have read GAMIN paper and examined their code very carefully. Kahla et al. [6] also suggest GAMIN is black-box.
>
>
> Table to compare GAMIN [a] and Ours:
> |  | **GAMIN [a]** | **Ours** |
> |-|-|-|
> |Training objective | Boundary-Equilibrium Adaptive loss [a] | Extended ACGAN loss |
> |Attack type | Black box | Label only |
> |Attack setting | Learn a surrogate for each target identity  | Learn a surrogate for all target identities |
> |Post-processing | Require (filtering etc.), see Fig D in attached PDF. Low-quality raw output | Not required |
> |Performance | Appear to be inadequate under black-box setting; 20% for MNIST [a] , 20% for CNN.  Note that the code of GAMIN has missing library, cannot run. See Fig D in PDF | Reasonable under more restricted label-only setting, e.g. 93.93% for CelebA, see our paper Table 3.|
>
> More details to compare GAMIN and ours:
>
> Central to GAMIN is the use of “Boundary-Equilibrium Adaptive loss” [a] as the training objective, which requires soft prediction from the target model $T$ to compute cross-entropy loss $L_H$. GAMIN's training objective:
>
> $L_S = L_H(y_t, Y_S) - k_t * L_H(y_t, Y_G)$
>
> See Fig. C in the attached PDF, where we highlight training objective in GAMIN algo.
> Note that soft prediction from target model $Y_S$ and $Y_G$ are required for cross-entropy $L_H$ of label $y_t$. The main idea behind GAMIN objective is to learn the distinction between noise inputs and generated samples.
>
> In our T-ACGAN, we develop an extended AC-GAN loss as training objective (our paper Eq. 4):
>
> $\mathcal{L}_{D,C} $
>
> $=  - E[\log P(s = Fake| x_{f})] - E[\log P(s = Real| x_p)]  - E[\log P(c = \tilde{y} | x_f)] $
>
> where $\tilde{y} = T(x_f)$.
> Our main idea is to use hard prediction of target model $T$ to obtain $\tilde{y}$ to learn the classifier head. Critically, no soft prediction from $T$ is needed.
>
> The difference between GAMIN's learning objective and ours is fundamental and significant.
>
> We remark that in their code, they replace cross-entropy loss with L1 loss, soft prediction is still required. We have submitted their code for reviewer easy reference, See Line 82,83 in trainer.py. See also Line 30, 31 in utils.py.
>
> [a] U. Aivodji et al. GAMIN: An Adversarial Approach to Black-Box Model Inversion.
>
> $
> $
> > What explains the difference in performance between ACGAN scenarios described in Section 6.2 and T-ACGAN given they seem to rely on   $D_{pub}$   which is obtained using the target model.
>
> The main reason is that for T-ACGAN we leverage diverse generated data to achieve rich decision knowledge transfer.
> Specifically, for ACGAN setups in Sec 6.2, $D_{pub}$ (or augmented $D_{pub}$) is labeled by target model $T$ for decision knowledge transfer to train the ACGAN. However, diversity of samples is limited by $D_{pub}$. Data augmentation only adds superficial variation.
> For our T-ACGAN, by integrating $T$ into ACGAN, generated samples are labeled by $T$ for decision knowledge transfer (Ln 146). The generated data randomly sampled from $G$ is very diverse, enabling transfer of rich decision knowledge. Furthermore, as training progresses, $G$ improves its conditional generation (see Fig. A.8 in Supp), achieving much class-balance decision knowledge transfer.
>
> $
> $
> >How does the intra-class diversity of the dataset affect the result of the attack?
>
> Reviewer's comment is insightful. First, we would like to emphasize that we follow strictly experiment setups in BREPMI [6].  Therefore, our experiment has already included private training data with considerable intra-class diversity, see Fig. E for some examples.
>
> We hypothesize that it becomes more challenging for all MI attacks (white-box, black-box, label-only) when intra-class diversity increases. It is because, with increased intra-class diversity, the variance of the feature distribution increases, and it becomes more difficult to reconstruct the identity features. However, the investigation needs much effort, e.g. to take into account the effect of reduced natural accuracy (model utility) due to increased intra-class diversity. We will look into this as future work as this is beyond the scope of this work ( new label-only attack).

---

> > ### Comment · Area_Chair_pwag · 2023-08-15
> > **Reviewer VBSE: Comparison with existing work**
> >
> > Dear Reviewer VBSE,
> >
> > Have the authors addressed the main issue of how their method compares to GAMIN and other comments provided in your review? Does their rebuttal change your assessment of the paper?

---

> > ### Comment · Reviewer_VBSE · 2023-08-16
> >
> >
> > Thank you for the clarifications. I suggest clarifying the limitation of the attack as well as adding the detailed comparison to GAMIN. I've increased my rating.

---

> > > ### Author Response · Authors · 2023-08-16
> > > **Thank you for the constructive and positive feedback**
> > >
> > > We thank Reviewer VBSE's constructive and positive feedback.
> > >
> > > We will follow Reviewer's advice and update the paper with detailed comparison with GAMIN (especially the fundamental difference in training objective and requirement of soft prediction) and limitation of label-only attack (up to 128x128 resolution for existing SOTA).
> > >
> > > If convenient, we appreciate if Reviewer could consider to review the initial assessment on Soundness, Presentation, Contribution. We appreciate Reviewer's valuable time. Thank you very much for all the constructive feedback.

---

### Author Rebuttal · Authors · 2023-08-10

We thank all the reviewers for their valuable time and effort to review our work. We appreciate the Reviewers' kind comments and recognition.

We would like to highlight that our paper focuses on Model Inversion, not Membership Inference as one of the reviewers suggested. Therefore, **comments of one of the Reviewers are completely irrelevant and out of context.**

 **Model Inversion and Membership Inference are totally different** as discussed in many works [a,b,c,d].

For model inversion, the objective is to **infer and reconstruct the training data of a trained model.**

For membership inference, the objective is to **determine whether a specific sample was used to train the model or not**.


In what follows, we provide comprehensive responses to all questions. We have provided anonymized link to Area Chair for the code of all additional experiments. We could provide more details if there are further questions. We hope that our responses can address the concerns and we sincerely hope that reviewers could consider increasing the ratings if our responses have addressed all the questions.


Thank you very much.


[a] Yuheng Zhang, Ruoxi Jia, Hengzhi Pei, Wenxiao Wang, Bo Li, Dawn Song. The Secret Revealer: Generative Model-Inversion Attacks Against Deep Neural Networks. CVPR-2020.

[b] Tanuwidjaja, H. C., Choi, R., Baek, S., & Kim, K. (2020). Privacy-preserving deep learning on machine learning as a service—a comprehensive survey. IEEE Access 2020.

[c] Rigaki, M., & Garcia, S. (2020). A survey of privacy attacks in machine learning. arXiv preprint arXiv:2007.07646.

[d] Yang Ziqi, Shao Bin, Xuan Bohan, Chang Ee-Chien, and Zhang Fan. 2020. Defending model inversion and membership inference attacks via prediction purification. arXiv preprint arXiv:2005.03915 (2020).

---

> ### Author Response · Authors · 2023-08-18
> **Thank You for the Thoughtful and Supportive Feedback**
>
> We sincerely thank Reviewers for their valuable feedback and thorough evaluation.
>
> In this paper, we propose a novel label-only model inversion (MI) attack, i.e., the challenging but practical scenario where the adversary only has access to the **hard label** prediction of the target model. Study of label-only model inversion attacks has been scarce. Our work provides new insights into the potential and performance of label-only model inversion attacks.
>
> In addition to the 37 experiments in the main paper and supplementary, our rebuttal further includes **13 new experiments** to respond to reviewers' feedback.  These experiments provide additional results on new target models and a large dataset, as well as testing on an additional defense MI model, higher resolution images, and an additional model architecture. We follow previous work strictly in the evaluation. In addition, we perform evaluation with **4 additional metrics**: user study, SSIM, average distance, and feature distance. **All the results consistently highlight the superior performance of our proposed label-only MI attack when compared to the state-of-the-art BREPMI.**
>
> **We have submitted all the code and pre-trained models of our experiments for reproducible research.**
>
> Thank you again for your valuable time and consideration. We hope Reviewers could consider to increase the ratings from initial decisions if our responses and additional experiments can resolve Reviewers' concerns. Thank you.
>
> $ $
>
> [BREPMI] Kahla, M., Chen, S., Just, H. A., & Jia, R. (2022). Label-only model inversion attacks via boundary repulsion. CVPR 2022.

---

> > ### Author Response · Authors · 2023-08-21
> > **Thank you for the positive and constructive feedback**
> >
> > Dear AC and Reviewers,
> >
> > We sincerely thank you for the positive feedback and recommendations.
> >
> > **Significance of our work**. We believe our work takes an important step in **Label-only Model Inversion (MI)** which is **the most challenging model inversion attack setting** where the adversary only has access to the **hard label** prediction of the target model. Investigation of label-only model inversion attacks has been scarce, with only prior work, BREPMI. Our proposed method provides superior Label-only MI Attack performance (>15% improvements across all MI benchmarks compared to BREPMI), new insights, and comprehensive analytical study into the potential of label-only MI attacks. Furthermore, **our work highlights rising privacy threats for machine learning models even under minimal information exposure**, i.e.: hard labels.
> >
> > $~$
> >
> > **Empirical validation and analytical justification**. We have conducted 50 experiments in total using 4 different datasets. To evaluate the results, we use 6 evaluation metrics including: Attack Accuracy, KNN distance, Feature distance, Average distance, SSIM, and Human Studies. **All the results consistently highlight the superior performance of our proposed label-only MI attack when compared to the state-of-the-art BREPMI.** We further conducted analysis to justify our approach (Sec 5 of main paper and Sec A of Supplementary). We also conducted, for the first time, 128x128 resolution Label-only MI experiments to further validate the efficacy of our proposed method (note that previous label-only MI study has experimented only up to 64x64).
> >
> > $~$
> >
> > **Research Reproducibility**. We have submitted all the code and the pre-trained models of our experiments for reproducible research. Our submission also includes a Google Colab demo (See Supp.) to facilitate hands-on interaction with our proposed method.
> >
> > $~$
> >
> > We sincerely thank you for your valuable time.
> >
> > $~$
> >
> > Sincerely,
> >
> > Authors.
> >
> > ------
> > [BREPMI] Kahla, M. et al. Label-only model inversion attacks via boundary repulsion. CVPR 2022.

---

### Decision · Program_Chairs · 2023-09-21

**Decision:**

Accept (poster)

**Comment:**

This paper proposes a method for label-only model-inversion (MI), i.e., where the adversary has only label access to the target model. The method uses transfer learning and generative models within a variation of the ACCGAN approach that uses access to the target model and the label that the target model generates. I thank the reviewers for engaging in the discussions and their authors for their extensive rebuttal. Overall, all the reviewers are positive about this paper being presented at NeurIPS and I agree.